



# MCSeg (v1.0): A Deep Learning Framework for Long-Term Large-Scale Mesoscale Convective Systems Identification and Precipitation Event Analysis

Peng Li [1], Zhanao Huang [1], Yongqiang Yu [2,3], Xi Wu [1], Xiaomeng Huang [4], and Xiaojie Li [1]

[1]Chengdu University of Information Technology, Chengdu, 610225, China
[2]Institute of Atmospheric Physics, Chinese Academy of Sciences, Beijing 100029, China
[3]University of Chinese Academy of Sciences, Beijing 100049, China
[4]Department of Earth System Science, Ministry of Education Key Laboratory for Earth System Modelling, Institute for Global Change Studies, Tsinghua University, Beijing, China

**Correspondence:** Xiaojie Li (lixj@cuit.edu.cn)

**Abstract.** Mesoscale Convective Systems (MCSs) are critical components of the climate system and are frequently responsible for extreme precipitation and other catastrophic weather events. Rapid and accurate identification of MCSs can significantly enhance our ability to respond to such extreme events. Traditionally, MCSs identification has relied on threshold-based methods, which are often limited by slower processing speeds and smaller detection areas. Recent advancements in deep learning

techniques for object recognition offer a promising alternative for MCSs identification. In this study, we propose an advanced approach to address the challenges associated with traditional threshold-based MCSs identification by creating a specialized dataset and training an MCSs recognition model. First, we constructed an MCSs identification dataset based on infrared satellite data, covering a spatial range (60°S - 60°N, 180°W - 180°E), and a temporal range from 2011 to 2023. Subsequently, by integrating a significance learning strategy and a multi-scale feature extraction method, we developed MCSeg, a novel MCSs

recognition model tailored specifically for mid- and low-latitude regions. Finally, we compared the MCSs identified using MCSeg with those identified using the threshold method and conducted precipitation event analysis. The results of the two methods showed a high degree of consistency, indicating the feasibility of applying deep learning methods to MCSs identification.

## 1 Introduction

Mesoscale Convective Systems (MCSs) are the primary drivers of highly destructive severe weather events, including heavy

rainstorms, hail, tornadoes, and flooding. Studies have shown that MCSs play a significant role in extreme precipitation events. For instance, between 1999 and 2003 in the United States, MCSs were associated with 66% of all extreme precipitation events and 74% of warm-season extreme precipitation events Schumacher and Johnson (2006). Similarly, in the East Asian Summer Monsoon Region, Ding et al. (2024) found that 91% of major floods and 87% of potential flood periods were linked to MCSs. Among these, 65% of major floods and 78% of potential flood periods were predominantly influenced by MCSs, while 38%

and 20% were driven by multiple types of MCSs, respectively. In 2023, China experienced 33 severe convective weather events, which affected over 1,100 counties with wind and hail disasters. These events impacted approximately 6.053 million people,





resulting in 57 fatalities, damaging 1,174.5 thousand hectares of crops, and causing direct economic losses of up to 11.73 billion yuan. Given their significant influence on weather, climate, and atmospheric environments, accurately monitoring the development and evolution of MCSs is of considerable importance for research in these areas.

Weather radar provides observations with high spatial and temporal resolution, making it well-suited for real-time identification and tracking of MCSs using classical storm-tracking algorithms such as SCIT Johnson et al. (1998) and TITAN Dixon and Wiener (1993). While these algorithms are effective for identifying small-scale supercells or isolated storms, they exhibit lower accuracy when applied to structurally complex MCSs. Additionally, the inherent limitations in radar observation range further constrain the identification scope of MCSs. Recent advancements in Earth-orbiting satellites have enabled the global

identification of MCSs using temperature thresholds and area coverage thresholds. Infrared satellite data at $11\mu m$ is commonly employed for MCSs identification Carvalho and Jones (2001); Laing et al. (2008). However, this approach often lacks detailed information on the internal structure of convection, such as precipitation distribution and vertical profiles. To address this limitation, an improved method utilizes precipitation data to enhance measurements of cloud cover and precipitation structure for MCSs identification Liu and Zipser (2013); Da Silva and Haerter (2023); Li et al. (2023). Both satellite-based methods can

accurately capture the life cycle of MCSs, including their initiation, maturation, and dissipation Liu and Zipser (2013); Roca et al. (2014); Laing and Michael Fritsch (1997). However, the thresholding method for MCSs identification in tropical and midlatitude regions is computationally intensive and time-consuming, requiring significant resources. This computational burden slows down downstream tasks such as tracking and forecasting, ultimately hindering the analysis of MCSs climatological characteristics. Therefore, an efficient automatic identification algorithm is needed to recognize a large range of MCSs.

Deep learning methods have been successfully applied to Camouflage Object Detection (COD) and Salient Object Detection (SOD) tasks in computer vision. These methods identify regions of interest by training end-to-end neural networks, offering fast inference speeds and low computational resource consumption. Leveraging the nonlinear fitting capabilities of neural network models can accelerate the recognition of MCSs and expand their identification range. However, morphological differences between mid-latitude and low-latitude regions make MCSs identification a hybrid task. Specifically, identifying MCSs in mid-

latitude regions resembles the COD task, while in low-latitude regions, it aligns more closely with the SOD task. Directly applying either SOD or COD methods alone to MCSs identification in these regions may lead to degraded model performance. Additionally, the significant scale variations of MCSs pose a challenge. During the extraction of large-scale MCSs, models may overlook the characteristics of small-scale MCSs, which has been confirmed to hinder the recognition effectiveness of deep learning models. This issue is particularly critical as small-scale MCSs are often ignored during the recognition process.

To address these challenges, we propose a deep learning model named MCSeg for identifying MCSs based on infrared satellite data. To handle the hybrid characteristics of MCSs in both mid-latitude and low-latitude regions, we introduce a significance learning strategy. This strategy allows MCSeg to independently learn nonlinear variations from disparate regions, enhancing the model's stability during training and improving recognition accuracy. Additionally, to tackle the issue of smallscale MCSs being overlooked during feature extraction, we design a multiscale semi-residual extractor. This extractor balances

the feature extraction of MCSs across different scales, ensuring that both large-scale and small-scale MCSs are adequately captured.



This paper has three main objectives. First, we propose a deep learning-based model for identifying MCSs in both tropical and mid-latitude regions. Second, we construct a comprehensive dataset for model training and testing. Third, we evaluate the effectiveness of the proposed method in identifying MCSs using available observations and perform a climatological

characterization of MCSs based on the identification results. The remainder of this paper is organized as follows. Section 2 reviews classical methods for MCSs identification. Section 3 describes the dataset used for training and evaluating the model. Section 4 details the proposed deep learning model. Section 5 presents both quantitative and qualitative results. Section 6 compares the effectiveness of our new method with the traditional thresholding method. Section 7 characterizes global MCSs and conducts a precipitation event analysis. Finally, Section 8 summarizes the research and suggests future research directions.

## 2   Related Work

### 2.1   MCSs identification

Long-term trends of MCSs and their relationship with precipitation have garnered increasing attention in regional studies Kunz et al. (2009); Masson and Frei (2016); Prein et al. (2017). In the United States, MCSs contribute more than 50% of annual and seasonal precipitation. For instance, in the Great Plains and Midwest regions, MCSs account for an average of

40% to 60% of total precipitation Feng et al. (2021). Furthermore, MCSs are responsible for the majority of warm-season rainfall in agriculturally significant plains and highland areas Haberlie and Ashley (2019). Similarly, extensive research has been conducted on the climatological characteristics of MCSs in East Asia and Europe. For example, Li et al. (2023) observed a 21.8% increase in MCS frequency and a 9.8% rise in intensity over the East Asian rainband between 2000 and 2020, leading to a greater contribution of MCSs-related precipitation to total rainfall. In the Asian monsoon region, MCSs frequency is

predominantly concentrated in tropical land and coastal areas, where strong lower-tropospheric water vapor flux convergence is common Zhai (2019). Using a combination of brightness temperatures and satellite precipitation estimates, Kukulies et al. (2021) identified MCSs occurrences within the tropical rainforest boundary (TPB) and surrounding low-elevation plains (LE) from 2000 to 2019. Additionally, Xian et al. (2024) investigated the evolution of MCSs using data from the Beijing High-Density Radar Wind Profile Relay Network (BHRWPRN) and Fengyun-2 geostationary satellites. Meanwhile, Da Silva and

Haerter (2023) utilized the latest Integrated Multi-Satellite Retrieval for Global Precipitation Measurement (IMERG) dataset to identify and track MCSs over 16 years in Europe, providing a comprehensive characterization of MCS climatology in the region.

With recent advancements in Earth-orbiting satellites, high-resolution cloud observations in both time and space now enable continuous global tracking of MCSs. A widely used approach involves leveraging infrared satellite data to identify MCSs based

on features such as lower cloud-top brightness temperatures Roca et al. (2014); Laing and Michael Fritsch (1997); Evans and Shemo (1996). For instance, Huang et al. (2018) applied temperature and area coverage thresholds using geostationary data to identify MCSs in the tropics (30°N–30°S) and integrated area overlap methods with Kalman filtering to track their evolution. However, infrared satellite data often lack detailed information on the internal structure of convection, such as precipitation distribution and vertical profiles. To address this limitation, another approach utilizes long-term satellite precipitation data to





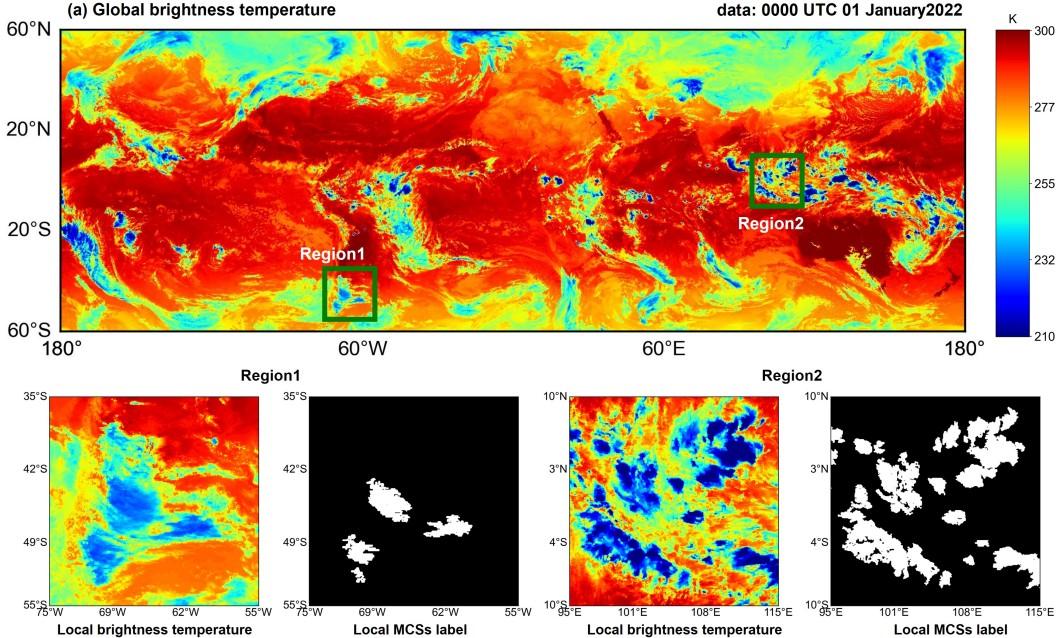

**Figure 1.** Visualization of BT data and MCS labels. **Region1** and **Region2** (highlighted in green boxes in (a)) correspond to the Camouflage Object Detection (COD) and Salient Object Detection (SOD) tasks, respectively.

enhance measurements of cloud cover and precipitation structure for MCSs identification Nesbitt et al. (2006); Liu and Zipser (2013); Da Silva and Haerter (2023); Li et al. (2023). For example, Hayden et al. (2021) identified and tracked MCSs within the IMERG precipitation field from 2014 to 2018, covering a spatial extent from 60°S to 60°N and designating contiguous areas greater than $1000km^2$ as MCSs. Similarly, Feng et al. (2021) successfully identified and tracked MCSs in both tropical and mid-latitude regions using IMERG precipitation data, bridging the gap in geostationary satellite coverage. Additionally, Rajagopal et al. (2023) employed the Forward in Time (FiT) algorithm Skok et al. (2013, 2010) to track MCSs in the IMERG precipitation field across the global tropics (30°N–30°S) over a decade (2011 to 2020). These studies have demonstrated that MCSs identification and tracking results derived from GPM and IMERG rainfall data align well with radar-based observations.

## 2.2 Why can deep learning techniques be used to identify MCSs?

From a computer vision perspective, MCSs recognition can be treated as an image binary classification task, which involves separating MCSs from brightness temperature (BT) data. We visualized the BT data in the MCSs recognition dataset, as shown in Figure 1. Obviously, the morphology of MCSs varies greatly between low and mid-latitudes. Specifically, supercooled clouds in low-latitude regions are characterized by their small size and well-defined profiles, typically corresponding to MCSs, as shown in Figure 1 (**Region2**). In contrast, cold clouds in mid-latitude regions exhibit bulky and indistinct profiles, with only the cold cloud cores meeting the temperature criteria for MCSs, as shown in Figure 1 (**Region1**). Therefore, identifying MCSs



in low-latitude regions can be viewed as identifying cold clouds, which is analogous to the SOD task. Conversely, identifying MCSs in mid-latitude regions can be regarded as identifying cold cloud cores, which is analogous to the COD task. The use of a single COD or SOD model to identify MCSs mid- and low- latitude can lead to over-identification and under-identification. Because the data input is completely random during the learning process of the model, each batch of data may contain data from both low and mid-latitude regions. This random data input process can lead to unstable model training.

Designing separate models for identifying MCSs in mid-latitude and low-latitude regions increases task complexity. There is a need to develop a unified model capable of recognizing MCSs across both regions, while addressing challenges such as over-recognition and under-recognition caused by significant scale variations and the dense distribution of MCSs.

## 3 Data

### 3.1 Data information

The Climate Data Record (CDR) combines observations from multiple satellites using the Goddard profile algorithm to provide global BT data derived from geostationary infrared satellites. This dataset is based on the three-hourly B1 data from the International Satellite Cloud Climatology Programme (ISCCP), with a spatial resolution of $0.07°$. Due to its high quality and global coverage, CDR has been widely used in convective identification studies, as demonstrated in Yang and Slingo (2001); Dias et al. (2012); Dong et al. (2016); Huang et al. (2018).

The Integrated Multi-satellite Retrievals for Global Precipitation Measurement (IMERG) mission provides the latest global precipitation data from the National Aeronautics and Space Administration (NASA). IMERG offers three data products: Early, Late, and Final runs, which cater to diverse application scenarios with varying latency times. By integrating precipitation retrievals from passive microwave (PMW) sensors on low Earth orbit (LEO) satellites and infrared (IR) sensors on geostationary satellites using the Goddard Profiling algorithm Kummerow et al. (2001, 2011, 2015), IMERG delivers high-quality precip-
itation estimates. In this study, we utilize the IMERG Final Precipitation L3 product, which has a temporal resolution of 30 minutes and a spatial resolution of $0.1°$.

### 3.2 Data processing

A strong correlation between low BT values and MCSs has been well-documented. MCSs identification is typically based on BT thresholds and minimum area coverage thresholds. Commonly, the temperature thresholds range from 208 K to 255
K Machado et al. (1998); Zhai (2019), while the area coverage thresholds range from 100 to $10000km^2$ Morel and Senesi (2002); Kolios and Feidas (2010). Implementing stricter criteria may exclude some valid MCSs, whereas adopting less stringent thresholds may result in the inclusion of spurious MCSs. Matthiasa (2011) reviewed existing criteria for MCSs identification and proposed a temperature threshold of 233 K and an area coverage threshold of $5000km^2$. These values were subsequently applied in studies by Fiolleau and Roca (2013) and Huang et al. (2018). Therefore, in this study, we adopt a temperature
threshold of 233 K and an area coverage threshold of $5000km^2$ to construct a more reliable dataset. Figure 2 illustrates the





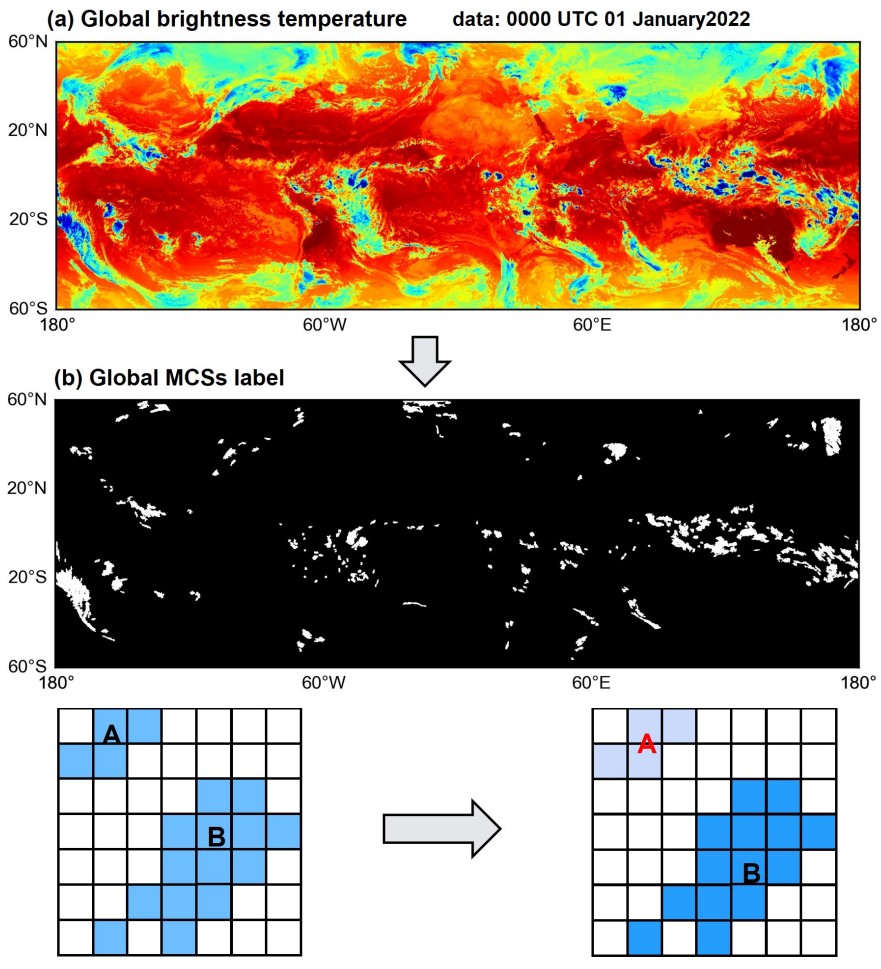

**Figure 2.** (a) bright temperature values from satellite data and (b) extracted MCSs. Lower part: description of the steps used to identify MCSs.

MCSs identification process using a simple example. First, BT values are extracted from satellite data, and the temperature threshold is applied to perform initial identification, yielding two potential MCSs (A and B). Next, regions that do not meet the area coverage threshold (e.g., A) are discarded. Regions satisfying both temperature and area coverage thresholds are labeled as MCSs (e.g., B).

## 3.3 MCSs dataset

We utilized BT data for the year 2022 and the period from March to August 2023 within the specified region (60°S–60°N, 180°W–180°E) to construct the training set and test set, respectively. The training and test sets consist of 2,920 and 1,472





BT data samples, respectively, with each BT data having a size of $1715 \times 5143$. During preprocessing, we identified a few extreme values in the BT data, including very large values ($> 400K$), very small values ($< 130K$), and null values. To ensure the stability of the model training process, these outliers were handled as follows: BT values $\geq 350$ K were set to 350 K, BT values $\leq 170$ K were set to 170 K, and null values were filled using bilinear interpolation. To fully utilize the features of the original infrared satellite imagery ($1715 \times 5143$), we expanded the edges of the images to ($2048 \times 5632$) and filled the additional regions with a BT value of 300 K (a non-MCSs value). The corresponding labels for these regions were set to 0. Each data was then divided into 44 sub-blocks of size $521 \times 512$. After preprocessing, the final training and test sets contained 128,480 and 64,768 samples, respectively.

We use max-min normalization to align the quantiles. By linearly transforming the original data, the comparison of data values before and after processing is balanced Henderi et al. (2021); Ahmed et al. (2022), thus speeding up the model convergence. The normalization process can be expressed as follows:

$$X = \frac{X - X_{min}}{X_{max} - X_{min}} \tag{1}$$

where $X$ represents the input data and $X_{max}$ and $X_{min}$ denote the maximum and minimum values of $X$, respectively.

## 4 Method

We propose MCSeg to identify MCSs in the global region (60°S–60°N, 180°W–180°E). It is a Convolutional Neural Network (CNN)-based model that primarily consists of four components: Input Projection, Encoder, Decoder, and Seg Head (see Figure 3 (a)). The Input Projection and Seg Head are responsible for initial feature extraction and final MCSs identification, respectively, both utilizing $3 \times 3$ convolutional kernels. In the Encoder, we introduce a multiscale semi-residual extractor (MSRE) to address the challenges of miss-segmentation in small-scale MCSs and boundary blurring in large-scale MCSs. The lightweight Decoder is employed to progressively restore the features extracted by the Encoder to their original resolution. To enhance the stability of the network during training, we adopt a significance learning strategy to balance the discrepancy between SOD and COD.

Given the input $X \in R^{H \times W \times 1}$ normalized by Eq. (1), this process is represented by the following equation:

$$\hat{Y} = f_\theta(X, \theta) \tag{2}$$

where $\hat{Y}$ denotes the identification results and $f_\theta(\cdot)$ is our module parameterized by $\theta$. We expect that $\hat{Y}$ is similar to the ground truth $Y$.

### 4.1 Multiscale Semi-residual Extractor (MSRE)

In the MCSs recognition task, the substantial variation in the scales of MCSs can lead to the loss of small-scale MCSs during feature extraction. Therefore, it is crucial to extract multi-scale features in this task. During the development of neural networks, numerous multiscale feature extraction models, such as VGG Simonyan and Zisserman (2014), ResNet He et al. (2016), and





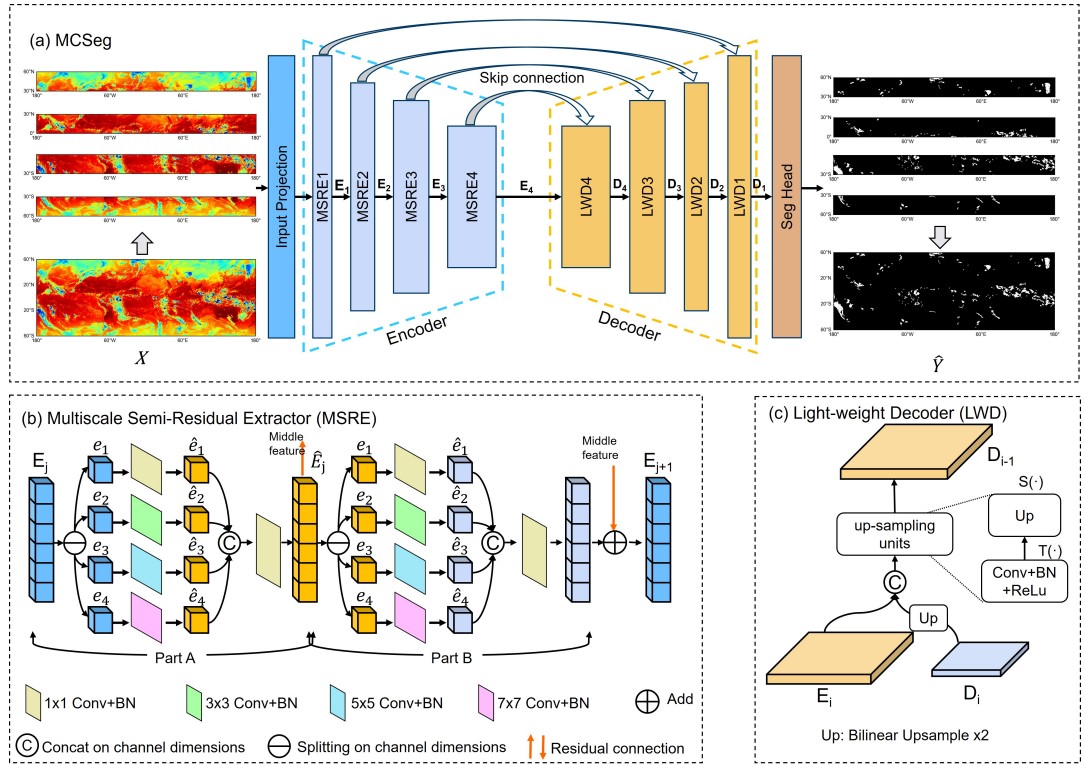

**Figure 3.** Sketch of the overall structure of (a) the MCSeg, (b) the Multiscale Semi-residual Extractor, and (c) the Lightweight Decoder (LWD).

MobileNet Howard (2017), have been proposed to address this issue by utilizing convolutional kernels of different sizes. Additionally, Transformer-based backbone networks, including ViT Vaswani (2017), PVT Wang et al. (2022), and SwinU Liu
et al. (2021), leverage attention mechanisms to capture global feature correlations. However, these networks often produce a large number of parameters, resulting in slower inference speeds.

The Multiscale Semi-Residual Extractor (MSRE) is proposed to balance multi-scale feature extraction and computational efficiency. As shown in Figure 3 (b), the MSRE consists of two parts: Part A and Part B. The primary difference between them is the absence of residual connections in Part A, whereas Part B incorporates residual connections. Below, we elaborate on the
design of Part A of the MSRE.

For the pyramidal feature $E_j$ generated by j-th layer of the Encoder, Part A will produce a new feature $\hat{E}_j$. Specifically, the feature map to Part A is initially divided into four blocks along the channel dimension. Each block $e_i$ is obtained by applying the split function $Split(\cdot) : R^{c \times H \times W} \Rightarrow R^{\frac{c}{4} \times H \times W}$ to the feature map $E_j$, as shown in Eq. (3).

$$e_i = Split(E_j) \tag{3}$$





where $c$ denotes the number of channels, $H$ and $W$ represent the height and width of the feature map, respectively. Compared to the input feature map, each sub-feature $e_i$ has the same spatial size, but with $1/4$ channels, $(i, j) \in \{1, 2, 3, 4\}$.

Afterwards, each of the four sub-features $(e_i)$ is processed by a different projection function to extract information at different scales. The transformed feature map $\hat{e}_i$ is obtained by applying the projection function $\psi_{\theta_i}(\cdot) : R^{\frac{c}{4} \times H \times W} \Rightarrow R^{\frac{c'}{4} \times H \times W}$, as defined in Eq. (4).

$$\hat{e}_i = \psi_{\theta_i}(e_i) \qquad (4)$$

where $\theta_i$ denotes the learnable parameters of $\psi_{\theta_i}(\cdot)$. For different $i$, the network employs $1 \times 1$, $3 \times 3$, $5 \times 5$, $7 \times 7$ convolutional layers followed by a sequential application of Batch Normalization and ReLU activation function. In the MSRE module, multi-scale processing is implemented in a piecewise manner, which not only facilitates the extraction of both global and local information but also significantly reduces the number of network parameters and accelerates model inference. However, the

transformed feature maps $\hat{e}_i$ are channel-wise independent due to non-overlapping slicing. To enable information exchange between these features and achieve better fusion of multi-scale information, we concatenate all $\hat{e}_i$ along the channel dimension and fuse the information using a $1 \times 1$ convolutional layer, followed by Batch Normalization and the ReLU activation function. This design allows the model to effectively capture both large-scale and small-scale MCS.

The above steps are repeated for the obtained $\hat{E}_j$ to generate the final output $E_{j+1}$ of the MSRE module. It is worth noting

that a residual connection is constructed between the inputs and outputs during this process. While residual connections are typically designed to mitigate the issue of vanishing gradients, applying residual design to all components (Part A and Part B) may lead to the propagation of redundant information. To address this, we adopt a semi-residual connection strategy, where residual connections are applied only in Part A but not in Part B. This approach ensures that the problem of vanishing gradients is effectively alleviated during model training, while simultaneously enabling the delivery of more refined MCSs information.

**4.2  Light-weight Decoder (LWD)**

The lightweight decoder (LWD) is employed to restore the resolution of the spatial features extracted by the Encoder. The detailed structure of the LWD is illustrated in Figure 3 (c). It comprises multiple up-sampling units, each of which includes a channel converter $T(\cdot)$ and a spatial restorer $S(\cdot)$. Given a pyramidal feature from the encoder $E_i \in R^{C_i \times 2H \times 2W}$ and a feature recovered by the decoder $D_i \in R^{C_i \times H \times W}$, the spatial restorer $S(\cdot)$ is applied to $D_i$ to align it with $E_i$ in the spatial dimension.

These features are then concatenated along the channel dimension and passed through the up-sampling units to produce the final output $D_{i-1} \in R^{\frac{C_i}{2} \times \sigma H \times \sigma W}$. The lightweight decoder at layer i can be formally expressed as:

$$D_{i-1} = S(T(Cat(S(D_i), E_i))) \qquad (5)$$

where $Cat(\cdot)$ denotes concatenation along the channel dimension. $T(\cdot)$ is implemented using a $1 \times 1$ convolution, followed by a sequential application of batch normalization and the ReLU activation function. $S(\cdot)$ represents the bilinear up-sampling

operator with a scale factor of $\sigma$.





## 4.3 Significance Learning Strategy

The climatological characteristics of mid-latitude and low-latitude MCSs exhibit significant differences within the region (60°S – 60°N, 180°W – 180°E). From the discussion in Section 2.2, it is evident that using a single COD or SOD model to identify MCSs across both low- and mid-latitude regions leads to model instability. To address this issue, we propose a Significance

Learning Strategy (SLS).

Specifically, SLS divides the data into significant and non-significant regions along the latitudinal direction, and then feeds these regions into the model for training. As shown in Figure 3 (a), the data is divided into four latitudinal regions. The first (60°N–30°N, 180°W–180°E) and fourth (60°S–30°S, 180°W–180°E) regions are located within the mid-latitude regions, while the second (30°N–0°N, 180°W–180°E) and third (30°S–0°S, 180°W–180°E) regions are situated in the low-latitude regions.

During training, the model learns data representations from one region at a time and then backpropagates them according to the gradient. By training on the first region, the model learns the data distribution characteristics relevant to the COD task. When the second region is introduced, the model shifts its focus towards learning the characteristics of the SOD task. Training on the third region further enhances the model's weighting parameters. Notably, after training on the second and third region, the model's learned distributions become more biased towards the SOD task, causing it to partially forget the distributions

learned for the COD task. With the introduction of the fourth region, the model's weights are reallocated towards the COD task. SLS ensures that each gradient update is task-specific, enhancing the model's stability during training and improving its effectiveness in identifying MCSs across both low- and mid-latitude regions. Furthermore, the SLS ensures that the model effectively mitigating the issue of catastrophic forgetting and preserving previously learned knowledge.

## 5 Experiments and Results

### 5.1 Implementation Details

MCSeg was implemented using PyTorch and trained on an NVIDIA RTX 4090 GPU. During training, the input data size was set to $512 \times 512$ , and no data augmentation techniques were applied. The batch size was set to 11, and the total number of training epochs was set to 10. The Adam optimizer was used with a learning rate of 0.001. To monitor the model's performance, we evaluated it on the test set after every 1/5 of the training data was processed. If the test metrics surpassed the current best

performance, the model parameters were saved; otherwise, they were discarded. This strategy ensured that only the optimal model parameters were retained. Additionally, early stopping was employed to terminate the training process if the test metrics did not improve over a predefined number of evaluations.

The loss function is employed to guide the model during the training process, ensuring that the model's outputs align more closely with the ground truth labels. To constrain MCSeg, we utilized two loss functions commonly adopted in classification

tasks: Dice loss (see Eq. 6) and Binary Cross-Entropy (BCE) loss (see Eq. 7).





$$\mathcal{L}_{Dice} = 1 - \frac{2 * |Y \cap \hat{Y}|}{|Y| + |\hat{Y}|} \tag{6}$$

$$\mathcal{L}_{BCE} = \frac{1}{N} \sum_{i=1}^{N} (Y \cdot log(\hat{Y}) + (1 - Y) \cdot log(1 - \hat{Y})) \tag{7}$$

where $Y$ and $\hat{Y}$ denote the labels and the predicted values of the model, respectively, and $N$ denotes the total number of samples.

## 5.2 Evaluation Metrics and Comparison Methods

We employed seven evaluation metrics, including mean absolute error ($M \downarrow$) Perazzi et al. (2012), maximal F-measure ($F_{\beta}^{mx} \uparrow$) Achanta et al. (2009), weighted F-measure ($F_{\beta}^{w} \uparrow$) Margolin et al. (2014), structural measure ($S_{\alpha} \uparrow$) Fan et al. (2017), mean enhanced alignment measure ($E_{\phi}^{m} \uparrow$) Fan et al. (2018), $Dice \uparrow$, and intersection over union ($IoU \uparrow$), to comprehensively assess the performance of MCSeg from multiple perspectives $M$ quantifies the average absolute difference between the predicted and ground truth results, where smaller values indicate higher accuracy. $F_{\beta}^{mx}$ represents the peak performance of the model by capturing the maximum F-measure value. $F_{\beta}^{w}$ provides a weighted balance between precision and recall, offering a comprehensive evaluation of the model's overall performance. $S_{\alpha}$ measures the structural similarity between the predicted and ground truth results, while $E_{\phi}^{m}$ evaluates the average enhanced alignment between them. $IoU$ and $Dice$ are statistical measures used to assess the overlap and similarity between the predicted and ground truth results, respectively. For all metrics except $M$, values closer to 1 indicate better performance.

We selected 10 comparative models designed for various segmentation tasks, which can be categorized into four groups: (1) dichotomous image segmentation methods, including FSANet Jiang et al. (2024) and BCMNet Cheng et al. (2023); (2) salient object detection (SOD) methods, including CPD Wu et al. (2019), F3Net Wei et al. (2020), and C2FNet Sun et al. (2021); (3) camouflaged object detection (COD) methods, including SINet Fan et al. (2021) and HitNet Hu et al. (2023); and (4) medical image segmentation methods, including PraNet Fan et al. (2020), SwinU Liu et al. (2021), and Poly-PVT Dong et al. (2021). All models were evaluated on the MCSs recognition dataset we constructed to compare the performance of these deep learning methods.

## 5.3 Quantitative and qualitative evaluation

We quantitatively evaluated MCSeg against other deep learning-based recognition models on the MCSs recognition task using seven widely adopted metrics, as summarized in TABLE 1. In the table, Our method (MCSeg) achieves strong performance across all seven evaluation metrics, significantly outperforming other deep learning methods and delivering results on par with SwinU. Moreover, our model is notably more compact, with parameter sizes of 16.2 MB for MCSeg compared to 114.76 MB for SwinU. Additionally, we provide Precision-Recall curves and F-measure curves to further evaluate the performance of



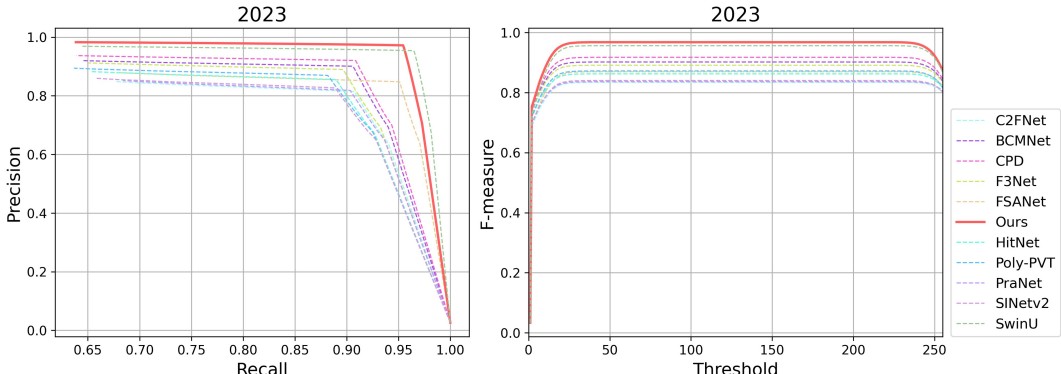

**Figure 4.** Precision-Recall (PR) curve (left) and F-measure curve (right) of the MCSeg and other methods are tested on the MCSs dataset.

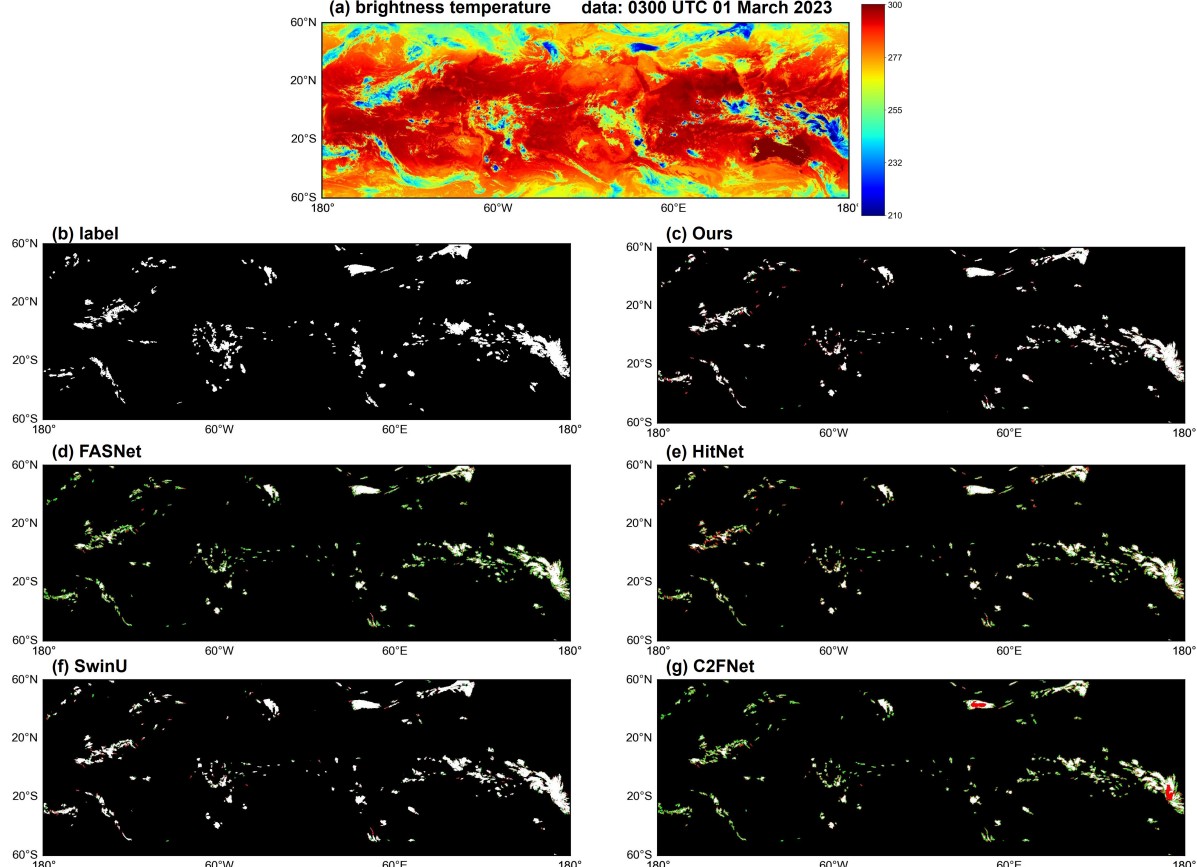

**Figure 5.** Visual comparison of MCS identification results using different methods in the study area. BT data corresponds to 0300 UTC on 01 March 2023. **Ours** represents the results from the proposed method, while **FASNet**, **HitNet**, **SwinU**, and **C2FNet** denote results from other deep learning methods. Green markers indicate over-identified regions, and red markers indicate under-identified regions.





**Figure 6.** Localized MCSs identification results of different methods. **Region1**, **Region2**, **Region3**, **Region4**, and **Region5** correspond to the green boxes in (a). BT data corresponds to 0300 UTC on 01 March 2023. Rows represent different methods, and columns represent different regions. **label** denotes the ground truth. Green markers indicate over-identified regions (present in **Ours** but not in **label**), while red markers indicate under-identified regions (present in **label** but not in **Ours**).





**Figure 7.** MCSs identification results for consecutive time periods using our proposed method in the study area. BT data spans from 0000 UTC to 2100 UTC on 02 March 2023.



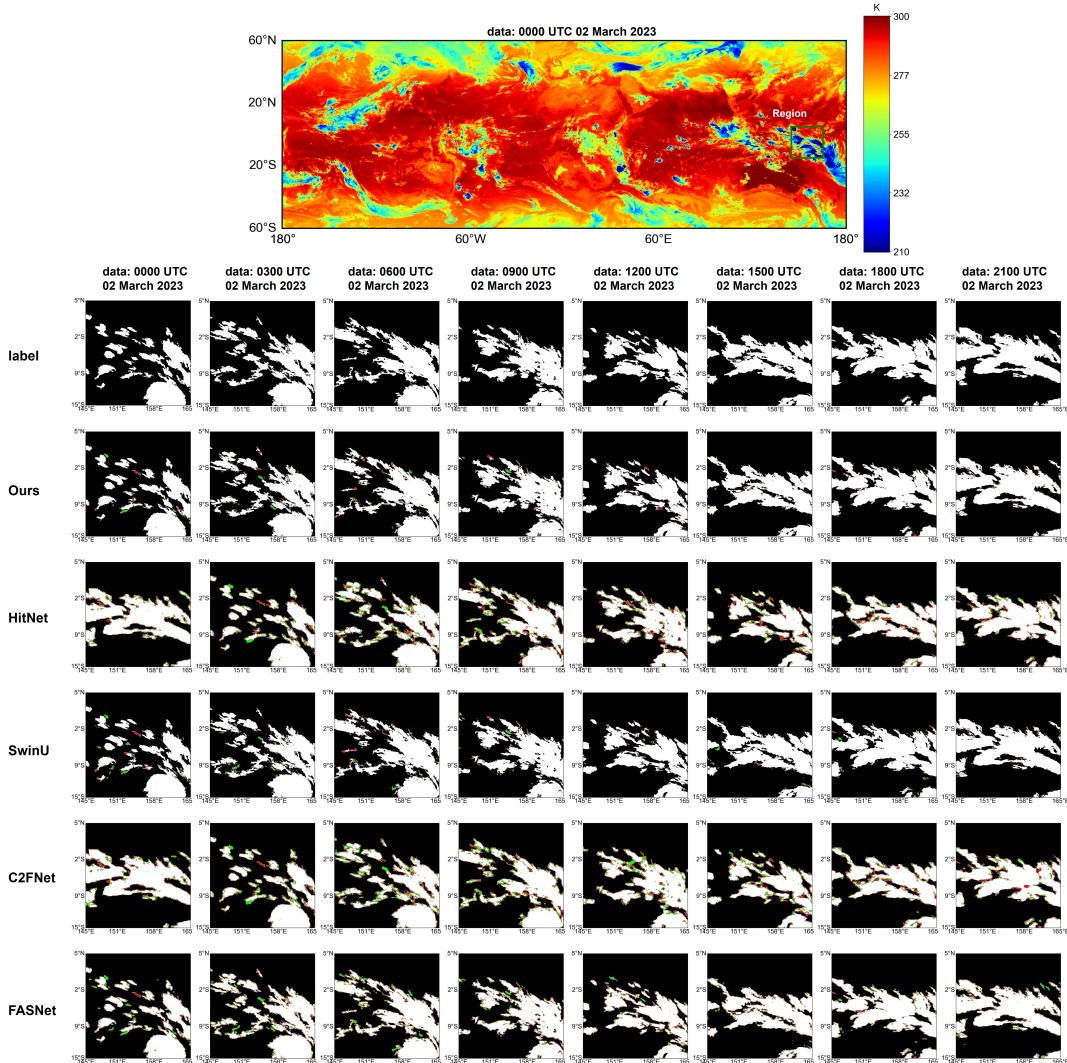

**Figure 8.** Localized MCS identification results of different methods across consecutive time periods. The regional scope corresponds to the green boxes in (a). BT data spans from 0000 UTC to 2100 UTC on 02 March 2023.





**Table 1.** Comparison of MCSeg with other methods for quantitative evaluation. Seven metrics ($M \downarrow$, $F_\beta^{mx} \uparrow$, $F_\beta^w \uparrow$, $S_\alpha \uparrow$, $E_\phi^m \uparrow$, $IoU \uparrow$, $Dice \uparrow$) were used to evaluate the performance of this methods. In this table, $\uparrow$ indicates that higher metrics are better and $\downarrow$ indicates that lower metrics are better. (All values in the table are multiplied by 10.)

| Metrics | CPD | PraNet | F3Net | SINet-v2 | Poly-PVT | C2FNet | SwinU | BCMNet | HitNet | FSANet | Ours |
|---|---|---|---|---|---|---|---|---|---|---|---|
| | | | | | Methods | | | | | | |
| $M \downarrow$ | 0.046 | 0.079 | 0.57 | 0.079 | 0.067 | 0.080 | 0.018 | 0.052 | 0.079 | 0.059 | **0.016** |
| $F_\beta^{mx} \uparrow$ | 9.176 | 8.911 | 8.911 | 8.402 | 8.722 | 8.340 | 9.563 | 8.340 | 8.402 | 8.694 | **9.680** |
| $E_\phi^m \uparrow$ | 9.855 | 9.852 | 9.852 | 9.815 | 9.844 | 9.784 | 9.864 | 9.784 | 9.815 | 9.788 | **9.866** |
| $S_\alpha \uparrow$ | 9.490 | 9.353 | 9.353 | 9.109 | 9.249 | 9.092 | 9.759 | 9.092 | 9.109 | 9.311 | **9.800** |
| $F_\beta^w \uparrow$ | 9.152 | 8.861 | 8.861 | 8.245 | 8.649 | 8.140 | 9.377 | 8.140 | 8.245 | 8.462 | **9.610** |
| $IoU \uparrow$ | 8.286 | 7.951 | 7.951 | 7.396 | 7.678 | 7.419 | 9.017 | 7.419 | 7.396 | 7.989 | **9.097** |
| $Dice \uparrow$ | 9.062 | 8.858 | 8.858 | 8.502 | 8.686 | 8.518 | 9.483 | 8.518 | 8.502 | 8.881 | **9.527** |

all methods on the MCSs recognition dataset, as illustrated in Figure 4. The red solid line, representing MCSeg, consistently
outperforms other models across most thresholds. The superior performance of MCSeg can be attributed to the well-designed
multiscale semi-residual encoder . The multiscale semi-residual encoder enhances the model's ability to extract multi-scale
features without increasing the number of parameters .

To visually compare the performance of different methods on the MCS recognition task, we plotted the recognition results
of MCSeg alongside other comparison methods, as shown in Figure 5. In this figure, white represents accurately recognized
regions, green denotes over-recognized regions (i.e., regions identified by **Ours** but not by **label**), and red indicates under-
recognized regions (i.e., regions identified by **label** but not by **Ours**). Overall, the MCSs identified by MCSeg and SwinU ex-
hibit the smallest deviations from the ground truth labels. In contrast, FSANet and HitNet show significant over-segmentation,
while C2FNet demonstrates more severe under-segmentation. We examined several localized regions within the green boxes
in Figure 6, which exhibit unique small-scale characteristics, and assessed the performance of various methods in identifying
MCSs at these specific locations. SwinU demonstrates clearer and more detailed recognition results; however, it suffers from
misidentification in edge regions due to the large scale variations and tight aggregation of MCSs. C2FNet and FASNet exhibit
limited recognition capabilities, resulting in significant over-segmentation and under-segmentation. HitNet produces smooth
edges but fails to accurately segment tightly aggregated MCSs. In contrast, MCSeg (Ours) shows high consistency with the
ground truth labels in terms of detail and outperforms the other methods. Additionally, MCSeg experiences fewer instances
of over-recognition and under-recognition, particularly in Region 2 and Region 5. Although we designed a specialized model
architecture to address the large scale differences of MCSs, it remains challenging to completely eliminate this issue.

Figure 7 illustrates the spatio-temporal identification performance of MCSeg, specifically its ability to identify MCSs over
consecutive time periods. The selected time period spans from 0000 UTC to 2100 UTC on 02 March 2023, and includes
eight frames of BT data. Overall, the results of MCS recognition over continuous time periods are consistent with those at
individual time points. Figure 8 compares the performance of MCSeg with other deep learning methods in recognizing MCSs
over continuous time periods. MCSeg (**Ours**) significantly reduces under-recognition and over-recognition, demonstrating





higher agreement with ground truth labels. Notably, SwinU performs better in recognizing edge regions of MCSs. However, all models struggle to accurately detect small-scale MCSs, leading to over-recognition in edge regions. The other models (C2FNet, HitNet, and FASNet) primarily detect the main body of MCSs but lack sufficient detail in edge regions, resulting in more severe under-recognition and over-recognition. Consequently, these methods face significant challenges in accurately localizing and identifying the content and morphological structure of MCSs in complex, time-continuous scenarios.

To validate the generalization ability of MCSeg, we conducted MCSs identification for the period from 2011 to 2021, and the results are presented in TABLE 2 and TABLE 3. Due to data quality issues in 2015, the results for that year are excluded. The validation results over the 10-year period, evaluated using the seven metrics, are consistent with those obtained from the test set, demonstrating the strong generalization capability of MCSeg. Notably, although the model was trained solely on data from 2022, it still achieves robust performance in identifying convective systems for 2011, despite the significant temporal gap. This indicates that MCSeg has effectively learned the feature distribution of MCSs, and its performance remains stable over time without degradation.

Both quantitatively and qualitatively, MCSeg demonstrates exceptional performance. The MCSs identified by MCSeg exhibit high consistency with those detected using the threshold-based method. By leveraging deep learning techniques, our method achieves a significantly faster recognition speed compared to the threshold-based approach. Specifically, while the threshold-based method requires 2 hours and 42 minutes to identify MCSs over a one-month time span, our method accomplishes the same task in just 1 minute and 40 seconds, representing a 200-fold improvement in identification speed (with variations depending on experimental hardware configurations).

## 6   Compared to the Threshold Method

In contrast to traditional threshold-based methods for identifying MCSs, including pure infrared (IR) methods and methods combining IR and precipitation, this study employs deep learning techniques to identify MCSs in low and middle latitudes (60°S–60°N, 180°W–180°E). In threshold-based approaches, MCSs are typically defined as cloud systems (i.e., continuous regions with BT values below a specified threshold and coverage areas exceeding a predefined threshold) that persist for several hours. These methods are effective for accurately identifying MCSs in tropical regions, where the cold cloud tops of most long-lived cloud systems reach mesoscale dimensions. To achieve more precise identification across different regions, researchers often adjust the thresholds accordingly. However, threshold-based methods are computationally expensive and exhibit slow identification speeds, particularly when applied to large-scale MCSs detection. In this study, we enhance the traditional threshold-based approach by leveraging deep learning techniques for MCSs recognition.

To demonstrate the superiority of our deep learning-based method (MCSeg) over the traditional threshold-based method for MCSs recognition, we compared the MCSs identified by both approaches in the US region. As illustrated in Figure 9, the results show a high degree of agreement between the MCSs detected by the two methods. Furthermore, we analyzed the relationship between the MCSs recognized by both methods and their associated precipitation. The results reveal that the precipitation proportions of MCSs identified by the two methods are nearly identical, with values ranging from 40% to 80% in both the



**Table 2.** Quantitative results of MCSeg and others on the 2011-2021 dataset, except for 2015. The best results are in bold. Our method outperforms the other methods in terms of metrics. (All values in the table are multiplied by 10.)

| | | Methods | | | | | | | | | | |
|---|---|---|---|---|---|---|---|---|---|---|---|---|
| Years | Metrics | CPD | PraNet | F3Net | SINet-v2 | Poly-PVT | C2FNet | SwinU | BCMNet | HitNet | FSANet | Ours |
| | $M \downarrow$ | 0.047 | 0.080 | 0.059 | 0.080 | 0.068 | 0.081 | 0.019 | 0.053 | 0.071 | 0.060 | **0.017** |
| | $F_\beta^{mx} \uparrow$ | 9.170 | 8.363 | 8.904 | 8.402 | 8.720 | 8.336 | 9.566 | 9.014 | 8.625 | 8.699 | **9.678** |
| | $E_\phi^m \uparrow$ | 9.855 | 9.797 | 9.851 | 9.815 | 9.843 | 9.784 | 9.864 | 9.853 | 9.839 | 9.790 | **9.866** |
| 2021 | $S_\alpha \uparrow$ | 9.481 | 9.094 | 9.344 | 9.101 | 9.240 | 9.083 | 9.757 | 9.408 | 9.205 | 9.307 | **9.797** |
| | $F_\beta^w \uparrow$ | 9.148 | 8.207 | 8.856 | 8.251 | 8.652 | 8.142 | 9.385 | 8.966 | 8.542 | 8.476 | **9.610** |
| | $IoU \uparrow$ | 8.273 | 7.426 | 7.941 | 7.390 | 7.667 | 7.410 | 9.017 | 8.098 | 7.615 | 7.989 | **9.092** |
| | $Dice \uparrow$ | 9.055 | 8.522 | 8.852 | 8.498 | 8.678 | 8.511 | 9.483 | 8.949 | 8.646 | 8.881 | **9.524** |
| | $M \downarrow$ | 0.045 | 0.077 | 0.056 | 0.077 | 0.066 | 0.078 | 0.018 | 0.051 | 0.068 | 0.058 | **0.016** |
| | $F_\beta^{mx} \uparrow$ | 9.175 | 8.368 | 8.909 | 8.407 | 8.724 | 8.340 | 9.565 | 9.019 | 8.630 | 8.703 | **9.679** |
| | $E_\phi^m \uparrow$ | 9.857 | 9.800 | 9.853 | 9.818 | 9.846 | 9.788 | 9.865 | 9.855 | 9.841 | 9.792 | **9.867** |
| 2020 | $S_\alpha \uparrow$ | 9.486 | 9.102 | 9.350 | 9.108 | 9.247 | 9.089 | 9.759 | 9.413 | 9.212 | 9.313 | **9.798** |
| | $F_\beta^w \uparrow$ | 9.153 | 8.213 | 8.861 | 8.256 | 8.656 | 8.145 | 9.383 | 8.972 | 8.546 | 8.478 | **9.611** |
| | $IoU \uparrow$ | 8.282 | 7.434 | 7.948 | 7.398 | 7.676 | 7.414 | 9.020 | 8.106 | 7.622 | 7.995 | **9.094** |
| | $Dice \uparrow$ | 9.060 | 8.527 | 8.856 | 8.503 | 8.684 | 8.514 | 9.485 | 8.953 | 8.650 | 8.885 | **9.525** |
| | $M \downarrow$ | 0.046 | 0.079 | 0.058 | 0.079 | 0.067 | 0.080 | 0.018 | 0.052 | 0.070 | 0.059 | **0.017** |
| | $F_\beta^{mx} \uparrow$ | 9.193 | 8.401 | 8.934 | 8.442 | 8.754 | 8.378 | 9.578 | 9.042 | 8.658 | 8.733 | **9.687** |
| | $E_\phi^m \uparrow$ | 9.857 | 9.802 | 9.853 | 9.819 | 9.845 | 9.790 | 9.866 | 9.855 | 9.841 | 9.795 | **9.868** |
| 2019 | $S_\alpha \uparrow$ | 9.495 | 9.115 | 9.360 | 9.121 | 9.258 | 9.106 | 9.765 | 9.423 | 9.223 | 9.326 | **9.802** |
| | $F_\beta^w \uparrow$ | 9.171 | 8.249 | 8.886 | 8.294 | 8.687 | 8.187 | 9.400 | 8.996 | 8.575 | 8.512 | **9.621** |
| | $IoU \uparrow$ | 8.317 | 7.477 | 7.986 | 7.441 | 7.718 | 7.466 | 9.044 | 8.142 | 7.664 | 8.034 | **9.114** |
| | $Dice \uparrow$ | 9.081 | 8.555 | 8.880 | 8.532 | 8.711 | 8.548 | 9.498 | 8.975 | 8.677 | 8.909 | **9.536** |
| | $M \downarrow$ | 0.047 | 0.080 | 0.059 | 0.080 | 0.068 | 0.082 | 0.018 | 0.053 | 0.071 | 0.060 | **0.017** |
| | $F_\beta^{mx} \uparrow$ | 9.193 | 8.399 | 8.932 | 8.439 | 8.751 | 8.369 | 9.578 | 9.041 | 8.656 | 8.733 | **9.688** |
| | $E_\phi^m \uparrow$ | 9.856 | 9.801 | 9.852 | 9.818 | 9.845 | 9.789 | 9.865 | 9.854 | 9.840 | 9.795 | **9.868** |
| 2018 | $S_\alpha \uparrow$ | 9.494 | 9.112 | 9.358 | 9.118 | 9.255 | 9.098 | 9.764 | 9.421 | 9.220 | 9.324 | **9.802** |
| | $F_\beta^w \uparrow$ | 9.171 | 8.246 | 8.886 | 8.292 | 8.685 | 8.179 | 9.402 | 8.996 | 8.575 | 8.515 | **9.623** |
| | $IoU \uparrow$ | 8.315 | 7.473 | 7.984 | 7.438 | 7.714 | 7.445 | 9.043 | 8.140 | 7.660 | 8.034 | **9.114** |
| | $Dice \uparrow$ | 9.080 | 8.553 | 8.879 | 8.530 | 8.709 | 8.534 | 9.497 | 8.974 | 8.674 | 8.909 | **9.536** |
| | $M \downarrow$ | 0.045 | 0.078 | 0.057 | 0.079 | 0.066 | 0.079 | 0.018 | 0.052 | 0.069 | 0.058 | **0.016** |
| | $F_\beta^{mx} \uparrow$ | 9.212 | 8.422 | 8.954 | 8.464 | 8.776 | 8.400 | 9.587 | 9.065 | 8.676 | 8.760 | **9.695** |
| | $E_\phi^m \uparrow$ | 9.857 | 9.804 | 9.853 | 9.821 | 9.846 | 9.794 | 9.866 | 9.856 | 9.842 | 9.799 | **9.869** |
| 2017 | $S_\alpha \uparrow$ | 9.508 | 9.128 | 9.373 | 9.135 | 9.271 | 9.119 | 9.771 | 9.435 | 9.234 | 9.342 | **9.808** |
| | $F_\beta^w \uparrow$ | 9.191 | 8.273 | 8.909 | 8.320 | 8.714 | 8.215 | 9.414 | 9.023 | 8.595 | 8.546 | **9.631** |
| | $IoU \uparrow$ | 8.347 | 7.501 | 8.015 | 7.467 | 7.744 | 7.484 | 9.059 | 8.169 | 7.686 | 8.068 | **9.127** |
| | $Dice \uparrow$ | 9.099 | 8.571 | 8.897 | 8.549 | 8.728 | 8.560 | 9.506 | 8.992 | 8.691 | 8.930 | **9.544** |





**Table 3.** Continuation of TABLE 2

| Years | Metrics | CPD | PraNet | F3Net | SINet-v2 | Poly-PVT | C2FNet | SwinU | BCMNet | HitNet | FSANet | Ours |
|---|---|---|---|---|---|---|---|---|---|---|---|---|
| | $M \downarrow$ | 0.045 | 0.078 | 0.056 | 0.078 | 0.066 | 0.079 | 0.017 | 0.051 | 0.069 | 0.057 | **0.016** |
| | $F_\beta^{mx} \uparrow$ | 9.230 | 8.457 | 8.976 | 8.499 | 8.803 | 8.429 | 9.598 | 9.087 | 8.706 | 8.791 | **9.702** |
| | $E_\phi^m \uparrow$ | 9.859 | 9.809 | 9.855 | 9.825 | 9.848 | 9.799 | 9.869 | 9.858 | 9.844 | 9.805 | **9.871** |
| 2016 | $S_\alpha \uparrow$ | 9.516 | 9.142 | 9.382 | 9.149 | 9.282 | 9.129 | 9.775 | 9.444 | 9.246 | 9.354 | **9.810** |
| | $F_\beta^w \uparrow$ | 9.208 | 8.311 | 8.931 | 8.359 | 8.742 | 8.247 | 9.430 | 9.047 | 8.628 | 8.582 | **9.640** |
| | $IoU \uparrow$ | 8.381 | 7.546 | 8.054 | 7.513 | 7.786 | 7.517 | 9.080 | 8.204 | 7.728 | 8.109 | **9.144** |
| | $Dice \uparrow$ | 9.119 | 8.600 | 8.921 | 8.579 | 8.755 | 8.582 | 9.518 | 9.013 | 8.718 | 8.955 | **9.553** |
| 2015 | - | - | - | - | - | - | . | - | - | - | - | - |
| | $M \downarrow$ | 0.043 | 0.076 | 0.055 | 0.076 | 0.064 | 0.078 | 0.017 | 0.050 | 0.067 | 0.055 | **0.016** |
| | $F_\beta^{mx} \uparrow$ | 9.254 | 8.495 | 9.006 | 8.538 | 8.836 | 8.464 | 9.599 | 9.118 | 8.735 | 8.832 | **9.699** |
| | $E_\phi^m \uparrow$ | 9.862 | 9.813 | 9.858 | 9.829 | 9.850 | 9.804 | 9.871 | 9.860 | 9.846 | 9.810 | **9.873** |
| 2014 | $S_\alpha \uparrow$ | 9.531 | 9.160 | 9.398 | 9.167 | 9.299 | 9.144 | 9.777 | 9.460 | 9.261 | 9.374 | **9.810** |
| | $F_\beta^w \uparrow$ | 9.226 | 8.348 | 8.957 | 8.398 | 8.771 | 8.284 | 9.429 | 9.075 | 8.654 | 8.623 | **9.634** |
| | $IoU \uparrow$ | 8.435 | 7.597 | 8.106 | 7.565 | 7.840 | 7.557 | 9.096 | 8.254 | 7.775 | 8.165 | **9.154** |
| | $Dice \uparrow$ | 9.151 | 8.633 | 8.953 | 8.613 | 8.788 | 8.608 | 9.526 | 9.043 | 8.748 | 8.989 | **9.558** |
| | $M \downarrow$ | 0.043 | 0.076 | 0.055 | 0.076 | 0.064 | 0.078 | 0.017 | 0.050 | 0.068 | 0.056 | **0.016** |
| | $F_\beta^{mx} \uparrow$ | 9.242 | 8.472 | 8.991 | 8.516 | 8.817 | 8.445 | 9.588 | 9.103 | 8.715 | 8.813 | **9.691** |
| | $E_\phi^m \uparrow$ | 9.861 | 9.811 | 9.857 | 9.827 | 9.849 | 9.801 | 9.869 | 9.859 | 9.845 | 9.807 | **9.872** |
| 2013 | $S_\alpha \uparrow$ | 9.526 | 9.151 | 9.392 | 9.158 | 9.292 | 9.138 | 9.772 | 9.453 | 9.252 | 9.366 | **9.806** |
| | $F_\beta^w \uparrow$ | 9.212 | 8.322 | 8.940 | 8.373 | 8.749 | 8.260 | 9.413 | 9.057 | 8.630 | 8.600 | **9.623** |
| | $IoU \uparrow$ | 8.414 | 7.567 | 8.082 | 7.536 | 7.812 | 7.537 | 9.078 | 8.230 | 7.746 | 8.141 | **9.139** |
| | $Dice \uparrow$ | 9.138 | 8.614 | 8.939 | 8.594 | 8.771 | 8.595 | 9.516 | 9.029 | 8.729 | 8.974 | **9.550** |
| | $M \downarrow$ | 0.043 | 0.076 | 0.055 | 0.076 | 0.064 | 0.077 | 0.017 | 0.050 | 0.067 | 0.055 | **0.016** |
| | $F_\beta^{mx} \uparrow$ | 9.246 | 8.474 | 8.994 | 8.518 | 8.820 | 8.451 | 9.589 | 9.108 | 8.718 | 8.817 | **9.692** |
| | $E_\phi^m \uparrow$ | 9.862 | 9.811 | 9.857 | 9.827 | 9.850 | 9.801 | 9.870 | 9.859 | 9.845 | 9.808 | **9.872** |
| 2012 | $S_\alpha \uparrow$ | 9.528 | 9.152 | 9.394 | 9.159 | 9.294 | 9.141 | 9.773 | 9.456 | 9.254 | 9.368 | **9.807** |
| | $F_\beta^w \uparrow$ | 9.217 | 8.325 | 8.944 | 8.375 | 8.755 | 8.268 | 9.416 | 9.065 | 8.635 | 8.605 | **9.626** |
| | $IoU \uparrow$ | 8.420 | 7.570 | 8.087 | 7.539 | 7.816 | 7.548 | 9.080 | 8.237 | 7.749 | 8.147 | **9.141** |
| | $Dice \uparrow$ | 9.142 | 8.616 | 8.942 | 8.596 | 8.774 | 8.602 | 9.518 | 9.033 | 8.731 | 8.978 | **9.551** |
| | $M \downarrow$ | 0.043 | 0.075 | 0.054 | 0.075 | 0.063 | 0.076 | 0.017 | 0.049 | 0.067 | 0.054 | **0.016** |
| | $F_\beta^{mx} \uparrow$ | 9.238 | 8.464 | 8.986 | 8.509 | 8.812 | 8.444 | 9.580 | 9.101 | 8.708 | 8.811 | **9.684** |
| | $E_\phi^m \uparrow$ | 9.862 | 9.811 | 9.857 | 9.827 | 9.850 | 9.801 | 9.869 | 9.860 | 9.846 | 9.808 | **9.872** |
| 2011 | $S_\alpha \uparrow$ | 9.527 | 9.150 | 9.392 | 9.158 | 9.292 | 9.142 | 9.770 | 9.454 | 9.251 | 9.368 | **9.803** |
| | $F_\beta^w \uparrow$ | 9.206 | 8.312 | 8.933 | 8.364 | 8.743 | 8.257 | 9.403 | 9.055 | 8.623 | 8.597 | **9.614** |
| | $IoU \uparrow$ | 8.409 | 7.557 | 8.076 | 7.526 | 7.803 | 7.541 | 9.068 | 8.225 | 7.735 | 8.138 | **9.129** |
| | $Dice \uparrow$ | 9.135 | 8.607 | 8.935 | 8.588 | 8.766 | 8.598 | 9.511 | 9.026 | 8.722 | 8.973 | **9.544** |





**Figure 9.** A comparison of the spatial distribution of warm season MCSs identified with the thresholding method (left) and the deep learning-based method (right) in 2021 is presented. The figures show the number of MCSs (a and e), the daily mean precipitation (b and f), the daily mean precipitation of MCSs (c and g), and the daily mean precipitation rate of MCSs (d and h). DL, deep learning; TM, threshold method.



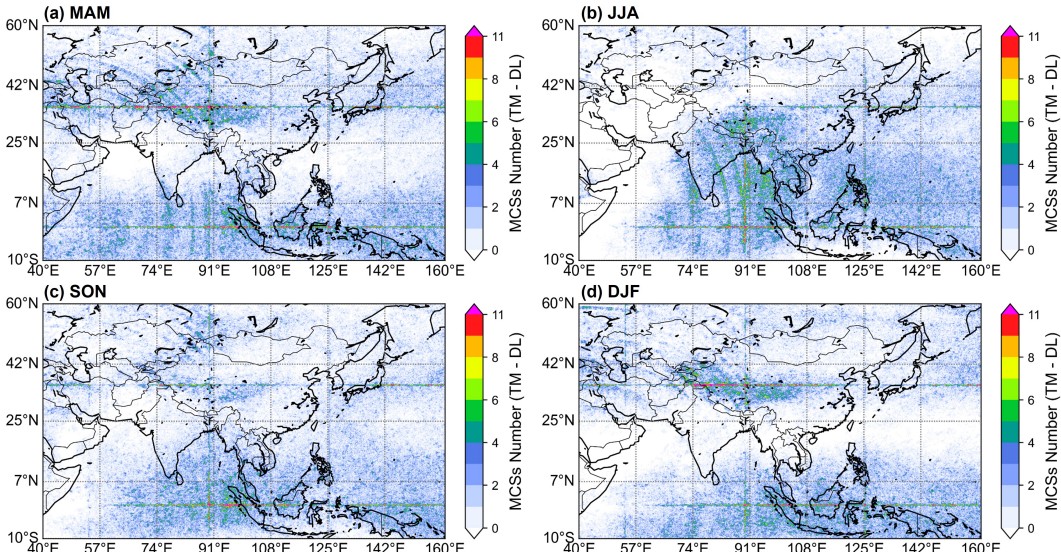

**Figure 10.** Differences in the mean spatial distribution of the number of MCSs identified based on the threshold method and the MCSeg method over the four seasons of 2018. (a) March to May, (b) June to August, (c) September to November, and (d) December to February.

Great Plains region and the southeastern coast. However, due to the slice learning strategy employed in MCSeg, horizontal or vertical stripes may occasionally appear at the junctions between slices.

  Figure 10 illustrates the spatial distribution of differences in the number of MCSs identified by the two methods across different seasons in Asia. The discrepancies between the results obtained by the two methods are relatively minor, with the overall difference in the number of MCSs ranging from 2 to 6. This demonstrates that our method achieves consistent MCSs

identification results compared to the traditional threshold-based approach. Notably, greater variations in the number of MCSs were observed at lower latitudes across the four seasons. During spring and winter, these differences were primarily concentrated over the Tibetan Plateau and the mid-latitudes to the north. In contrast, during summer and autumn, the disparities were mainly located near the Bay of Bengal and over the western Pacific warm pool. Additionally, vertical and horizontal bands are visible in the results, which can be attributed to the slice training strategy employed during the model training process. The

regions with significant disparities correspond to the boundaries of these slices.

  These analyses, conducted exclusively in Asia and the Americas, demonstrate that our methodology aligns closely with threshold-based approaches in identifying MCSs. In terms of computational efficiency, our method requires fewer resources and less time compared to threshold-based methods. This novel algorithm is applicable across all seasons in both tropical and mid-latitude regions, enabling the rapid development of a comprehensive database for MCS identification at low and

mid-latitudes.



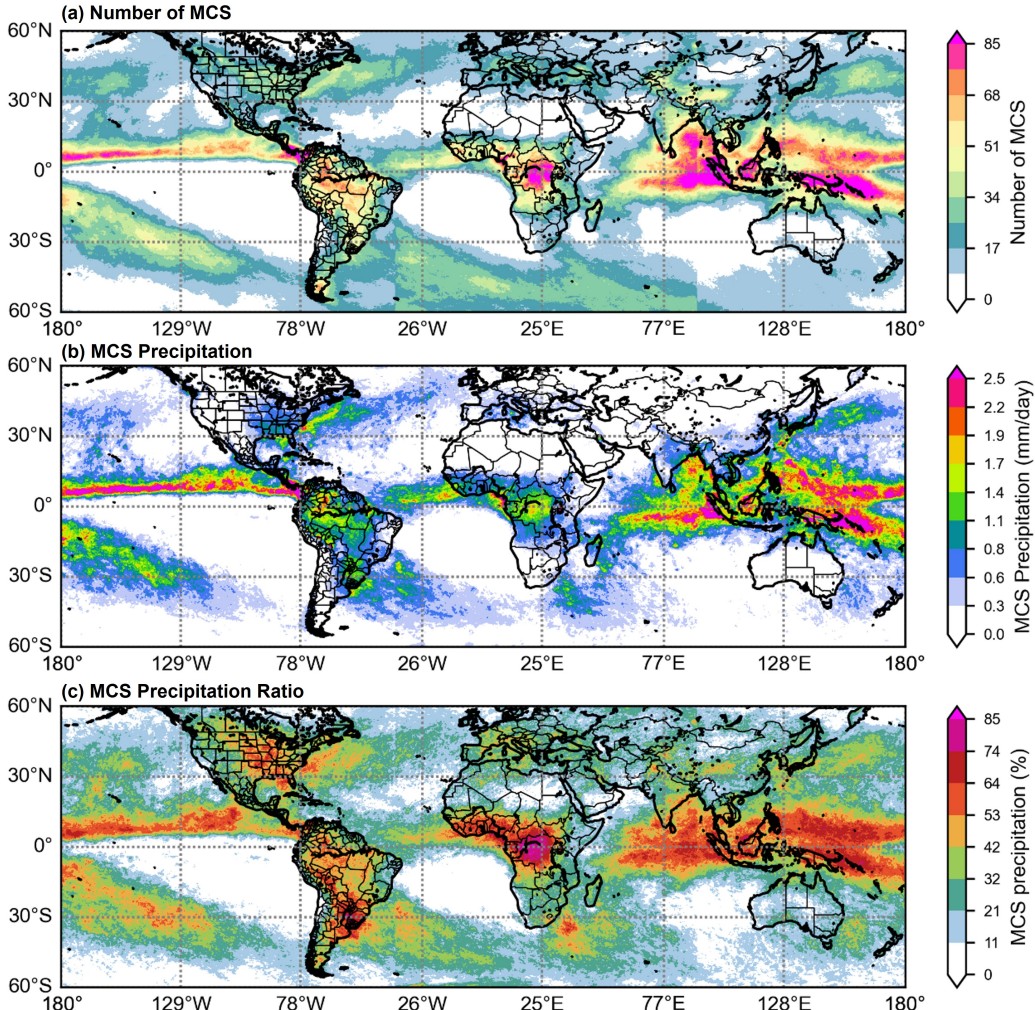

**Figure 11.** Distribution of MCSs (a), MCS precipitation (b), and the contribution of MCSs to total precipitation (c) in the mid- and low-latitude regions in 2018.



**Figure 12.** Contribution of MCSs to total precipitation in the mid- and low-latitude regions in 2018 for all seasons. (a) March to May, (b) June to August, (c) September to November, and (d) December to February.



## 7 Global MCSs Characteristics

The global distribution of the average annual number of MCSs, MCSs-related precipitation, and the contribution of MCSs to total annual precipitation in 2018 is illustrated in Figure 11. The annual number of MCSs is derived by aggregating the values of each pixel on the MCSs grid. MCSs-related precipitation is calculated by extracting precipitation values from the grid using

the MCSs mask. The contribution of MCSs to total annual precipitation is determined by dividing MCSs-related precipitation by total precipitation. As shown in Figure 11(a) and (b), the occurrence of MCSs is strongly correlated with precipitation patterns. MCSs are prevalent in both tropical and mid-latitude regions. In the tropics (30°S–30°N, 180°W–180°E), MCSs are particularly frequent in Central Africa, the Indo-Pacific warm pool, and the eastern Pacific, with an average annual occurrence exceeding 80 events. High MCSs activity is also observed in the Intertropical Convergence Zone (ITCZ) and monsoon troughs.

Conversely, MCSs frequency is notably lower in the tropical southeastern Pacific Ocean and the southern Atlantic Ocean, where stable atmospheric conditions suppress convective development. The spatial distribution of tropical MCSs identified by MCSeg is consistent with other MCSs datasets, including those derived from mesoscale convective complexes Laing and Michael Fritsch (1997), deep convective weather states in the tropics Tan et al. (2015), and precipitation characteristics Yuan and Houze (2010). This agreement further validates the applicability of our method in tropical regions. In mid-latitude regions,

MCSs predominantly occur in areas such as the Great Plains of the United States, the Tibetan Plateau, and eastern China. Over oceanic regions, MCSs are frequently observed along the coasts of the Americas and Central Africa, as well as in the Indian Ocean, largely due to the influence of warm ocean currents.

The distribution of MCSs-related precipitation is generally consistent with the spatial density of MCSs. Regions with the highest mean annual MCSs-related precipitation include the eastern tropical Pacific off the coast of Colombia, the coastal

waters west of Nigeria, the Indo-Pacific warm pool, and the seas surrounding Papua New Guinea. In most tropical regions, the contribution of MCSs to total annual precipitation exceeds 50%, reaching over 70% in areas such as the waters off Colombia, the Bay of Bengal, and the Indo-Pacific warm pool. Yuan and Houze (2010) utilized BT and precipitation data to identify mature MCSs and reported that tropical MCSs contribute approximately 56% of total precipitation, which aligns well with the findings of this study. Among continental regions, Argentina in South America, the Great Plains of North America, and Central

Africa exhibit the highest contributions of MCSs-related precipitation, accounting for more than 70% of total precipitation. This is followed by regions such as Europe, Northwest China, and India, where MCSs contribute approximately 40-60% of total precipitation. These results are supported by previous studies: Roca et al. (2014) analyzed MCSs over a single season (June to September) and found that MCSs contribute 40-60% of tropical rainfall, while Nesbitt et al. (2006) demonstrated that MCSs dominate tropical precipitation, contributing 60-70% of total rainfall across the tropics. The spatial distribution of MCSs

precipitation contributions identified in this study is consistent with findings from previous studies employing threshold-based approaches, further validating the robustness of our methodology.

As illustrated in Figure 12, the contribution of MCSs to regional precipitation exhibits strong seasonal variability across many regions of the globe. However, in the Indo-Pacific warm pool region, the proportion of precipitation attributable to MCSs remains consistently above 50% throughout the year. During the summer months (June-July-August, JJA), MCSs contribute



more than 70% of total rainfall in regions such as the central United States, the eastern coast of China, central Africa, and the Bay of Bengal. In autumn (September-October-November, SON) and winter (December-January-February, DJF), MCSs play a dominant role in the continental regions of South America, accounting for over 50% of total precipitation. In Africa, the spatial distribution of MCSs-related precipitation shifts southward during autumn and winter compared to summer. During summer, mid-latitude regions in the Northern Hemisphere generally experience lower MCSs-related precipitation, with the exception of

the US Great Plains. In contrast, mid-latitude regions in the Southern Hemisphere exhibit a higher proportion of MCSs-related precipitation during this season. This pattern reverses during winter, with Northern Hemisphere mid-latitudes experiencing increased MCSs activity while Southern Hemisphere mid-latitudes show a decline.

## 8    Conclusion

In this study, a novel methodology was developed to rapidly and accurately identify global (60°S–60°N, 180°W–180°E) long-

term (2011–2023) Mesoscale Convective Systems (MCSs) using deep learning techniques. In this study, a training set and a test set for MCSs identification were constructed. Besides, the MCSs recognition model (MCSeg) was developed. Experimental results demonstrated that MCSeg outperformed existing deep learning-based recognition methods, as confirmed by both qualitative and quantitative evaluations. To further validate the model's generalizability, MCSeg was tested on BT data from 2011 to 2021 (excluding 2015). The validation results showed that MCSeg maintained robust performance across these datasets,

with no significant degradation in accuracy. This indicates that MCSeg effectively learned the distribution of MCSs and could accurately identify MCSs in both mid- and low-latitude regions. Moreover, the model's ability to identification future MCSs was reinforced by its consistent performance over extended historical periods.

To assess the practical applicability of MCSeg, its results were compared with those obtained using a threshold-based method in the Great Plains region of the United States during the warm season (March–August) of 2021. The comparison

revealed high consistency between the two methods in terms of the number of MCSs, MCSs-related precipitation, and the ratio of MCSs precipitation to total precipitation. Additionally, the discrepancy between the number of MCSs identified by MCSeg and the threshold method was analyzed across four seasons in Asia in 2018. While regions with significant variations in MCSs counts shifted with seasonal transitions, the discrepancy between the two methods remained consistent, ranging from 4 to 6 MCSs per season. The implementation of the slice training strategy led to a notable increase in this discrepancy at the slice

level.

Finally, a precipitation analysis was conducted on the MCSeg identification results for low- and mid-latitude regions. The spatial distribution of MCSs in 2018 was analyzed in terms of MCSs-related precipitation and the contribution of MCSs to total precipitation. The results were consistent with other datasets, such as those derived from Mesoscale Convective Complexes, tropical deep convective weather states, and rainfall characteristics, further validating the applicability of our method.

In summary, the deep learning model demonstrated high accuracy in MCSs recognition, comparable to the threshold method but significantly faster. However, the training set used for the model was still derived from threshold-based methods, introducing uncertainties that may affect the model's recognition results. In short, the MCSs datasets produced using thresholding will lead

to diversity in the recognition results of the model. This study has several limitations. For instance, only a single-threshold MCSs identification dataset was generated to evaluate the performance of MCSeg. Additionally, the study primarily focuses
on MCSs recognition and does not address the classification of MCSs types. In future work, unsupervised deep learning models could be explored for MCSs recognition. Such models would eliminate the need for threshold-based MCSs labeling, thereby avoiding the subjectivity associated with threshold setting. Furthermore, regionalized MCSs identification models with high spatial and temporal resolution could be developed by integrating multi-source data, such as precipitation data and radar reflectivity data. Another promising direction is to classify MCSs based on their characteristics and analyze the relationships
between different types of MCSs and various extreme weather events.

This study represents a preliminary application of deep learning techniques to MCSs recognition. Future research could extend this approach to track MCSs trajectories and perform spatio-temporal predictions of MCSs using deep learning methods.

*Code availability.* The source code used in this study is available at https://doi.org/10.5281/zenodo.17078318. The specific test and training datasets used in the experiment are publicly available at https://doi.org/10.5281/zenodo.17077599.

*Data availability.* The GPM IMERG precipitation data V06 are obtained from the NASA Goddard Earth Sciences Data and Information Services Center (https://disc.gsfc.nasa.gov/datasets/). The Geostationary IR Channel Brightness Temperature (BT)- GridSat-B1 Climate Data Record (CDR) are obtained from the National Centers for Environmental Information (https://www.ncei.noaa.gov/products/gridded-geostationary-brightness-temperature).

*Author contributions.* PL: investigation, methodology, writing(original draft),writing(review and editing). ZH: validation, writing(review
and editing), visualization. XL, XH: resources, supervision, funding acquisition. XW, YY: supervision, validation.

*Competing interests.* Xiaomeng Huang is a member of the editorial board of GMD.

*Acknowledgements.* This work was supported by the Sichuan Science and Technology program (Grant No. 2024YFG0001), and the National Natural Science Foundation of China (Grant Nos. 42130608, 42125503 and 42430602).



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
