# Peer review of "MCSeg (v1.0): A Deep Learning Framework for Long-Term Large-Scale Mesoscale Convective Systems Identification and Precipitation Event Analysis"

_EGUsphere, 2025_

## Referee Comment (RC1)

**Review for**

**MCSeg (v1.0): A Deep Learning Framework for Long-Term Large-Scale Mesoscale Convective Systems Identification and Precipitation Event Analysis**

**by Peng Li et al.**

**Summary:**

The study by Li et al. presents a new algorithm to identify Mesoscale Convective Systems (MCS) that is based on machine learning. The performance of the new method is compared to different other methods, including the threshold-based traditional one. It is shown that the new algorithm is able to clearly identify MCS in the tropical and tropical regions, and that its computational performance (time to identify MCS) is superior to other methods, and in particular to the traditional one.

The presentation is mostly clear, the figures well supporting the statements. Still, at some places I wonder whether all information really has to be shown in the manuscript, and if the focus on the algorithmic aspects remain clear enough towards the end. Furthermore, I especially wonder why the new methods is so much faster than the traditional one.

I think the paper becomes publishable if the concerns listed below are adequately addressed. I don't think that further analysis has to be done, but that the text needs to be clarified at some places. Given this, I would recommend something between minor and major revisions are needed.

**Specific comments:**

1. I see that the new algorithm is well able to identify MCS, as it compares well to the threshold-based approach in case studies and also reproduces well global MCS climatologies. What I do, however, not really understand is why the new algorithm is that much faster than the threshold-based algorithm. In fact, I would argue that there is computationally a simpler approach than applying threshold to an input field, as is done for the traditional approach based on brightness temperature and precipitation. As mentioned in the text, the difference is very large: about 3 hours for the traditional approach, but only 2 minutes for the new approach. Is this really only for the identification of the MCS, or does the difference come from tracking the MCS and so attributing to the single MCS time-continuous labels? Possibly, I miss an essential point in the discussion?! It would be helpful to discuss in greater detail the reason for this large difference in computation cost?

As a specific point: In the introduction (L37-39) it is explicitly written that the traditionl identification of MCS hinders the analysis of MCS climatological characteristics. Is this really true? If we rely on 30 min Imerg and BT data from, say, 2003 to present, that would be 20 years of data that has to be processed. Even with moderate computational resources that should be feasible in a reasonable amount of time and thus does not hinder climatological analysis. In fact, such global analysis have already been done.

2. The authors convincingly show that their new algorithm performs very well compared to other ones, both in accuracy of MCS identification and in computational cost. This is shown in case study figures, in global climatologies and also in tables. I wonder whether all details are actually needed in the manuscript. So, for example, I can imagine that table 2 and its continuation table 3 are too detailed

and at least part of them could be provided in supplementary material. Possibly, the same applies to part of figures 6-8.

3. In Section 7, some global characteristics of MCS are listed, e.g., their frequency and link to precipitation. The previous sections of the paper discuss algorithmic aspects of the MCS identification, or they compare the new algorithm's performance with other ones. This section, however, is much more strongly focused on meteorology and it also states that many of the findings are already known from existing literature. I think that this meteorological discussion only partly fits into the overall 'structure' of the manuscript. So, either the meteorological analysis should be extended and so bring new insights that can be gained based on the new, more efficient (faster) MCS identification. Since this is not the main focus of the study, I instead suggest to 'frame' this section also more strongly to show that the new method is able to reproduce existing climatologies of MCS.

4. There are some aspects that need to be clarified or improved in paper structure:

- Parts of the abstract are rather technical, at least if one is not too familiar with the machine learning approaches. As an example: many readers will not immediately understand what a 'significance learning strategy' and/or a 'multi-scale feature extraction methjod' is. Are these pieces of information really necessary in the abstract?

- L147: The edges of the image are expanded and filled with non-MCS values of brightness temperature. Does this mean that the domain is not periodic in zonal direction, and if so: how does this lead to artificial MCS artefacts near the dateline?

- L133: Here, it is mentioned that 'spurious MCS' are to be excluded. What are spurious MCS, and if they are still MCS, why should they be excluded?

- COD vs SOD: I am not completely sure whether I understand the distinction between the two? In Figure 1, it is written that the MCS in tropical and extratropical regions (Region 1 and 2 in the figure) are different in structure? Is it that the MCS differ in their degree of spatial clustering, or do all the individual MCS differ in their structure? For readers less familiar with the distinction between tropical and extratropical MCS some further background information (and references) could be helpful?

- Most likely related to the previous point: The title of Section 2.2 is somewhat 'misleading'. I would not have doubted that machine learning can be used to identify MCS. The real point of Section 2.2 is that the authors suggest to use different approaches (SOD vs. COD) for low and mid latitudes. This should be reflected also in the title of the Section 2.2.

- L67-76: This part is less about MCS identification, but more about background information (occurrence frequency, link to precipitation). I think it would better fit into the introduction or, possibly, the discussion.

---

## Referee Comment (RC2)

**Review of egusphere-2025-3622**

**Title:** *MCSeg (v1.0): A Deep Learning Framework for Long-Term Large-Scale Mesoscale Convective Systems Identification and Precipitation Event Analysis*

**Aurhors:** *Peng Li, Zhanao Huang, Yongqiang Yu, Xi Wu, Xiaomeng Huang, and Xiaojie Li*

The manuscript describes a deep learning (DL)-based methodology to identify Mesoscale Convective Systems (MCS). The methodology accounts for one single algorithm with different approaches for mid- and low-latitude regions. The performances of the algorithm are compared with other machine learning (ML)-based approaches and with a physical-based approach. The analysis shows comparable results between the new developed algorithm and the physical-based approach, while a general outperformance with respect to the other ML approaches.

The paper is well organized and well written. Although I appreciate the work done by the authors with the development of one single algorithm for both medium and low latitudes, I do not see any particular improvement for the scientific community. As I stated below in the specific comment, an ideal case with perfectly working DL algorithm can exactly reproduce the training and test dataset. In the specific case, the only reason to consider the present work as an added value is the computing time. But, I do not see how a simple threshold-based method can be 200 times slower than a DL-based method (hours vs less than two minutes).

- the characteristics of the MCS have to be reported. Not all the readers are familiar with this precipitating structure.

- line 17 and many others: in sentences like this the correct way to report the reference is to put it all in parentheses: (Schumacher and Jonson, 2006).

- lines 125-126: the use of IMERG final product is only related to the analysis of the precipitation distribution linked to the MCSs carried out in Section 6 and 7? It is not totally clear from the text.

- lines 137-139: explain better the area coverage threshold. It is not clear whether there must be spatial continuity up to reach 5000 km$^2$ or not. This affect also the interpretation of the results. When you show, in Figure 6 or 8 for instance, the green and read areas, how these affect the identification of a MCS? I mean, is there a threshold in number of pixels (and consequently in extension area) to say if a given structure is or not a MCS?

- lines 146-148: what the reason to increase so much the image size? I can understand that at the edges of your image you can suffer of padding, but in Section 4 you mention you use a 3x3 convolutional kernel.

- Table 1 and 2: I do not understand the choice of multiplying the coefficient by 10. Also because at lines 259-260 you state that all metrics, except M, have 1 as perfect score.

- lines 309-314: really the application of a threshold on 240 images (8 images per day for 30 days in a month) requires almost 3 hours? Honestly, it is hard to trust on this. On the other hand, this does not imply that your approach cannot be faster than the threshold-based one. In addition, Section 5 describes the comparison between MCSeg and other DL-based methods, but your conclusion mainly focus on the shorter computational time with respect to the threshold-based method. Another weak point is the choice of the 10 DL-based methods. At line 261 you state, "We selected 10 comparative

models designed for various segmentation tasks…". Basically, you are comparing an MCS oriented method with other "generic" methods. In my opinion, the comparison is not fair.

- lines 323-324: I disagree with this sentence. If your model was perfect, it would be able to exactly reproduce the training (and the test, supposing perfect generalization capabilities) data. This means that you are not able to enhance the threshold-based method but, at most, to equal it (since your dataset is built exploiting the threshold-based method).

- the last part of the paper (Figure 9-12 and Section 7) could be considered a bit out of topic with respect to the rest of the paper. The analysis is useful (no doubt), but cannot be considered a consequence of the development of the MCSeg algorithm. In particular, the results shown in Figure 9 highlight negligible differences between MCSeg and threshold-based algorithms. In addition, vertical and horizontal bands (lines 338-339 – I would add curving bands as well) present in Figure 10 have to be fixed. A data post-processing can be applied in order to remove this issue.

---

## Editor Comment (EC1)

**Review:** egusphere-2025-3622

**Title:** MCSeg (v1.0): A Deep Learning Framework for Long-Term Large-Scale Mesoscale Convective Systems Identification and Precipitation Event Analysis

**Recommendation: Major Revision**

**Summary**

This study develops a machine-learning based approach to detect MCSs trained on a referenced dataset produced by a traditional Tb threshold-based method. The authors compared their results against other generic ML-based methods to show their algorithm performs better than generic ML methods, and it performs substantially faster than the traditional methods. Some climatological comparisons of MCS statistics were also performed against the reference dataset over the U.S. and Asia to show agreements.

I think it is a worthwhile effort to explore ML techniques as an alternative to traditional physically-based methods to identify MCSs. One obvious advantage is the computational efficiency of ML-based methods, as the study demonstrates. Unsupervised ML-based methods trained on existing reliable MCS datasets that can reproduce salient features of the physically-based MCS tracking algorithms offers the community new tools to study MCSs. To that end, I support such efforts to be pursued and published.

However, there are several major issues in the current study that prevents me from recommending publication at the current stage:

- The traditional Tb-only MCS identification method used as reference in this study has been shown to overestimate MCSs in the mid-latitude because large cold clouds can be produced by different weather systems other than MCSs (e.g., extratropical cyclones, fronts), particularly during the cold and transition seasons. Recent studies have addressed some of those biases by incorporating precipitation data along with Tb to reduce false MCS identification in the mid-latitudes (Feng et al. 2021; Prein et al. 2024). The authors have cited some of these studies, but did not pursue such more advanced methodology to produce reference/training datasets for their ML approach.
- There are several global MCS tracking datasets available (Feng et al. 2021; Prein et al. 2023; Rajagopal et al. 2023), some used both Tb and precipitation data to detect MCSs (Feng et al. 2021; Prein et al. 2023). The authors should compare their results directly with these established datasets to quantify the performance of the ML approach. In addition, recent studies have compared multiple MCS tracking algorithms and documented their impacts on MCS statistics (Prein et al. 2024; Feng et al. 2025). These studies should be referenced and discussed in the context of the choice of the reference dataset used.
- One of the overlooked aspects of MCS identification in this study is the temporal dimension. Besides identifying a cloud system with low Tb and large area, physically-based MCS algorithms also require *persistence* of the cloud systems meeting the size (area) and intensity (Tb) criteria (i.e., systems must maintain the size and intensity for longer than several hours). Further, traditional tracking algorithms

connect the individual cloud systems in time to obtain lifecycle information for each system, thus providing information of their initiation location, timing, growth rates, movement and trajectories. These aspects are critically important to understanding the mechanisms of MCS development (e.g., Roca et al. 2017; Elsaesser et al. 2022; Chen et al. 2023; Barton et al. 2025), and is also used to perform process evaluations of MCSs in numerical models (e.g., Zhang et al. 2021; Dong et al. 2023, 2025; Feng et al. 2023; Prein et al. 2024; Cui et al. 2024). Based on what was presented, it does not look like the ML method provide such temporal evolution of individual systems, which is a severe drawback compared to traditional methods. The authors should discuss this limitation, explain why it is not considered, and whether it would be pursued in future works.

- Because the ML method also did not train on a dataset that already include temporal information of MCSs, the identification purely based on snapshots may differ substantially from established tracking datasets. I strongly recommend the authors compare their ML-based MCS dataset with one of those established datasets mentioned above. In fact, one of the coauthors have developed long-term global MCS tracking dataset before (Huang et al. 2018), why is that not used for the training?

In addition, the motivation of developing an ML-based method could be further strengthened. Currently, the only argument why an ML method is superior is computational performance. However, majority of the applications for MCS tracking algorithms are in research, where high computational efficiency is welcomed but not a deal breaker. The authors argue their approach could be used in real-time monitoring of MCSs, but it is not clear to me what actual advantage would such an algorithm provide in operational forecasting. I do see a potential application to research though, because virtually all existing MCS algorithms require reasonably high temporal resolution to track MCSs (i.e., no less than 3 hourly), this often hinders applying these traditional tracking algorithms to model outputs that do not provide sufficient temporal resolution data, e.g., HighResMIP (Haarsma et al. 2016). If an ML-based method *trained on tracked MCS data* can accurately identify MCSs based only on snapshots, that will allow it to be applied to datasets with insufficient temporal outputs and yet still reliably identify MCSs, hence achieving a goal that traditional methods cannot.

**Additional comments**

1. Section 3.l, Dataset details: the authors did not mention the time resolution of the ISCCP dataset and also did not provide which version of the IMERG data was used. They also did not mention how the IMERG dataset was matched with the ISCCP data since they have different spatiotemporal resolutions.
2. Evaluation issues:
   a. It is unclear why only specific years/periods were selected to validate the performance in different regions: U.S. (Mar-Aug 2021), Asia (all seasons in 2018), global (all seasons in 2021). Why not consistently evaluate the climatology of all years used in the study (2011-2023) for more robust statistics?
   b. Fig. 6: the global scale is too small to see details of individual MCSs, only the largest (coldest) clouds are visible.

c. Fig. 7: How is the number of MCSs calculated? Given that no tracking in time is performed. How is individual MCS objects per time step aggregated to number of systems? Also, the exact months should be listed in the caption as "warm season" is ambiguous.

d. Fig. 8: larger different in the cold season over Tibetan Plateau may be related to non-MCSs misidentified based on Tb-only. There are some ring-like artifacts not mentioned (seem to have boundary ~90°E), is that related to stitching artifacts of Tb between two geostationary satellites in the ISCCP data?

e. Fig. 9: why only validate for 1 year when the study include data from 2011-2023? The global results should be directly compared with established MCS datasets as I mentioned in my major comments. There are also large discontinuities of MCS numbers at ~30°W and ~90°E that were not discussed.

**References**

Barton, E. J., Klein, C., Taylor, C. M., Marsham, J., Parker, D. J., Maybee, B., et al. (2025). Soil moisture gradients strengthen mesoscale convective systems by increasing wind shear. Nature Geoscience. https://doi.org/10.1038/s41561-025-01666-8

Chen, X., Leung, L. R., Feng, Z., & Yang, Q. (2023). Environmental Controls on MCS Lifetime Rainfall Over Tropical Oceans. Geophysical Research Letters, 50(15), e2023GL103267. https://doi.org/10.1029/2023GL103267

Cui, W., Galarneau Jr., T. J., & Hoogewind, K. A. (2024). Changes in Mesoscale Convective System Precipitation Structures in Response to a Warming Climate. Journal of Geophysical Research: Atmospheres, 129(9), e2023JD039920. https://doi.org/10.1029/2023JD039920

Dong, W., Zhao, M., Ming, Y., Krasting, J. P., & Ramaswamy, V. (2023). Simulation of United States Mesoscale Convective Systems using GFDL's New High-Resolution General Circulation Model. Journal of Climate, 36(19), 6967-6990. https://doi.org/10.1175/JCLI-D-22-0529.1

Dong, W., Zhao, M., Guo, H., Harris, L., Cheng, K.-y., Zhou, L., & Ramaswamy, V. (2025). Comparison of Global Mesoscale Convective System Simulations in a Global Storm-Resolving Model and a High-Resolution General Circulation Model. Journal of Climate, 38(10), 2339-2356. https://doi.org/10.1175/JCLI-D-24-0303.1

Elsaesser, G. S., Roca, R., Fiolleau, T., Del Genio, A. D., & Wu, J. (2022). A Simple Model for Tropical Convective Cloud Shield Area Growth and Decay Rates Informed by Geostationary IR, GPM, and Aqua/AIRS Satellite Data. Journal of Geophysical Research: Atmospheres, 127(10), e2021JD035599. https://agupubs.onlinelibrary.wiley.com/doi/abs/10.1029/2021JD035599

Feng, Z., Leung, L. R., Hardin, J., Terai, C. R., Song, F., & Caldwell, P. (2023). Mesoscale Convective Systems in DYAMOND Global Convection-Permitting Simulations. Geophysical Research Letters, 50(4), e2022GL102603. https://doi.org/10.1029/2022GL102603

Feng, Z., Prein, A. F., Kukulies, J., Fiolleau, T., Jones, W. K., Maybee, B., et al. (2025). Mesoscale Convective Systems Tracking Method Intercomparison (MCSMIP): Application to DYAMOND Global km-Scale Simulations. Journal of Geophysical Research: Atmospheres, 130(8), e2024JD042204. https://doi.org/10.1029/2024JD042204

Haarsma, R. J., Roberts, M. J., Vidale, P. L., Senior, C. A., Bellucci, A., Bao, Q., et al. (2016). High Resolution Model Intercomparison Project (HighResMIP v1.0) for CMIP6. Geosci. Model Dev., 9(11), 4185-4208. https://doi.org/10.5194/gmd-9-4185-2016

Prein, A. F., Mooney, P. A., & Done, J. M. (2023). The Multi-Scale Interactions of Atmospheric Phenomenon in Mean and Extreme Precipitation. Earth's Future, 11(11), e2023EF003534. https://doi.org/10.1029/2023EF003534

Prein, A. F., Feng, Z., Fiolleau, T., Moon, Z. L., Núñez Ocasio, K. M., Kukulies, J., et al. (2024). Km-Scale Simulations of Mesoscale Convective Systems Over South America—A Feature Tracker Intercomparison. Journal of Geophysical Research: Atmospheres, 129(8), e2023JD040254. https://doi.org/10.1029/2023JD040254

Roca, R., Fiolleau, T., & Bouniol, D. (2017). A Simple Model of the Life Cycle of Mesoscale Convective Systems Cloud Shield in the Tropics. Journal of Climate, 30(11), 4283-4298. https://journals.ametsoc.org/doi/abs/10.1175/JCLI-D-16-0556.1

Rajagopal, M., Russell, J., Skok, G., & Zipser, E. (2023). Tracking Mesoscale Convective Systems in IMERG and Regional Variability of Their Properties in the Tropics. Journal of Geophysical Research: Atmospheres, 128(24), e2023JD038563. https://doi.org/10.1029/2023JD038563

---

## Author Comment (AC2)

**Reply to Referee comment 1:**

We extend our sincere gratitude to the reviewer for the time and effort dedicated to evaluating our manuscript, as well as for the invaluable comments and constructive suggestions provided. Below, we provide a point-by-point response to each comment and concern. All revisions made to the manuscript text are presented in green font for clarity.

1. I see that the new algorithm is well able to identify MCS, as it compares well to the threshold-based approach in case studies and also reproduces well global MCS climatologies. What I do, however, not really understand is why the new algorithm is that much faster than the threshold-based algorithm. In fact, I would argue that there is computationally a simpler approach than applying threshold to an input field, as is done for the traditional approach based on brightness temperature and precipitation. As mentioned in the text, the difference is very large: about 3 hours for the traditional approach, but only 2 minutes for the new approach. Is this really only for the identification of the MCS, or does the difference come from tracking the MCS and so attributing to the single MCS time-continuous labels? Possibly, I miss an essential point in the discussion?! It would be helpful to discuss in greater detail the reason for this large difference in computation cost?

Response: We thank the reviewer for this insightful question regarding the computational efficiency of our method. The dramatic speedup of MCSeg stems from fundamental differences in computational paradigms, which we clarify below.

1. Clarification of the Comparison Benchmark

We first clarify that the reported times (approximately 3 hours for the traditional method versus 2 minutes for MCSeg) represent a fair comparison of the MCS identification stage only, conducted on the same hardware, and do not include any subsequent tracking steps. This ensures an equitable comparison.

2. The Hidden Costs of the Traditional Threshold Method

The reviewer is correct that a single threshold operation is computationally cheap. However, a complete and robust traditional MCSs identification pipeline involves a series of computationally intensive post-processing steps:

- Connected Component Labeling (CCL): This constitutes the primary performance bottleneck. The algorithm must scan the entire two-dimensional global grid at each timestep to identify and label all independent, contiguous cloud clusters. The computational complexity of this operation scales with the number of grid points, and its inherent sequential data dependencies make it notoriously difficult to parallelize efficiently.
- Morphological Filtering: Subsequently, each labeled candidate cloud cluster must undergo a series of filters based on physical definitions . This requires expensive feature extraction on thousands of irregularly shaped objects, operations that are largely performed serially.
- 3. Sources of Efficiency in the Deep Learning Model In contrast, the efficiency of MCSeg, as an end-to-end deep learning model, derives from several key factors:
- Highly Parallelized Forward Pass: Model inference is essentially a process of data flowing through

a fixed sequence of layers. The core operations are transformed into highly regular tensor operations, which are perfectly suited for massive parallelization across the thousands of computing cores available on a GPU.

- Integrated Identification Pipeline: The model implicitly and synergistically performs feature extraction, contextual understanding, and pixel classification within a single, compact forward pass. It bypasses the necessity for the separate, sequential CCL and complex morphological filtering pipeline required by the traditional approach.
- Hardware-Level Optimization: The entire inference process is built upon deep learning frameworks optimized for accelerators like GPUs. The underlying computational libraries are extremely optimized for these fundamental operations, far surpassing the performance of handwritten, sequential CPU code.

In summary, the speed advantage does not come from a simplification of the algorithmic logic, but from replacing an inherently sequential algorithm pipeline with complex post-processing with a highly parallelized, hardware-friendly, and integrated model.

As a specific point: In the introduction (L37-39) it is explicitly written that the traditionl identification of MCS hinders the analysis of MCS climatological characteristics. Is this really true? If we rely on 30 min Imerg and BT data from, say, 2003 to present, that would be 20 years of data that has to be processed. Even with moderate computational resources that should be feasible in a reasonable amount of time and thus does not hinder climatological analysis. In fact, such global analysis have already been done.

Response: We thank the reviewer for this comment and for rightly pointing out the overstatement in our original wording. The reviewer is correct that traditional threshold-based methods have been successfully employed to produce valuable global MCSs climatologies, as evidenced by several key studies in the field. Our intention was to highlight the significant computational burden and inefficiency of these methods, which, while not rendering climatological analysis impossible, does pose a substantial practical barrier to rapid, iterative, and large-scale analysis. To accurately reflect this and to incorporate the reviewer's valid point, we have revised the relevant sentences in the introduction.

2. The authors convincingly show that their new algorithm performs very well compared to other ones, both in accuracy of MCS identification and in computational cost. This is shown in case study figures, in global climatologies and also in tables. I wonder whether all details are actually needed in the manuscript. So, for example, I can imagine that table 2 and its continuation table 3 are too detailed and at least part of them could be provided in supplementary material. Possibly, the same applies to part of figures 6-8.

Response: We thank the reviewer for the suggestion regarding the presentation of our detailed results. We agree that streamlining the main manuscript in this manner enhances its readability for a broad audience, while still ensuring that comprehensive data remain available to interested researchers. In accordance with this advice, we have relocated the extensive experimental results from Tables 2 and 3 to the Supplementary Information. This allows the main text to focus on the high-level conclusions drawn from these comparisons without being encumbered by the full datasets. Furthermore, we have

carefully assessed Figures 5-8. While we believe a curated selection of these panels is essential in the main text to visually support our central claims regarding identification accuracy, we have moved the more specific regional analysis into the Supplementary Information, retaining only the global-scale visualization results in the main body. All relocated items are explicitly cited in the main text. We believe these changes have significantly improved the focus and presentation of our work and thank the reviewer again for this constructive recommendation.

3. In Section 7, some global characteristics of MCS are listed, e.g., their frequency and link to precipitation. The previous sections of the paper discuss algorithmic aspects of the MCS identification, or they compare the new algorithm's performance with other ones. This section, however, is much more strongly focused on meteorology and it also states that many of the findings are already known from existing literature. I think that this meteorological discussion only partly fits into the overall 'structure' of the manuscript. So, either the meteorological analysis should be extended and so bring new insights that can be gained based on the new, more efficient (faster) MCS identification. Since this is not the main focus of the study, I instead suggest to 'frame' this section also more strongly to show that the new method is able to reproduce existing climatologies of MCS.

Response: We sincerely thank the reviewer for this observation. We agree that the primary focus of the manuscript should remain on the validation of our novel deep learning algorithm. Following the reviewer's excellent suggestion, we have completely restructured the manuscript to reframe the climatological analysis as a key component of the algorithm's validation. Specifically, we have merged the original Section 6 ("Compared to the Threshold Method") and Section 7 ("Global MCSs Characteristics") into a single, comprehensive new section titled "Comprehensive Validation of the MCSeg Algorithm". Within this new section: The regional comparisons with the threshold-based method are presented as the first tier of validation. The global-scale analysis is now explicitly positioned as the second tier of validation. Instead of presenting meteorological findings, we now explicitly state that the purpose is to verify whether our algorithm can successfully reproduce well-established climatological patterns from the literature (e.g., spatial distributions, precipitation contributions, and seasonal cycles). We consistently use language that frames the agreement with prior studies as direct evidence of our algorithm's accuracy and physical realism.

- 4. There are some aspects that need to be clarified or improved in paper structure:
- Parts of the abstract are rather technical, at least if one is not too familiar with the machine learning approaches. As an example: many readers will not immediately understand what a 'significance learning strategy' and/or a 'multi-scale feature extraction methjod' is. Are these pieces of information really necessary in the abstract?

Response: We thank the reviewer for this suggestion. Following the reviewer's advice, we have revised the abstract to replace the technical terms 'significance learning strategy' and 'multi-scale feature extraction method' with more general descriptions of their functions.

- L147: The edges of the image are expanded and filled with non-MCS values of brightness temperature. Does this mean that the domain is not periodic in zonal direction, and if so: how does

**this lead to artificial MCS artefacts near the dateline?**

Response: The reviewer rightly pointed out that our initial preprocessing approach, which involved expanding image edges and filling them with a constant BT value of 300 K, did not account for zonal periodicity and could indeed introduce artificial boundaries along the dateline. In response, we have revised the data preprocessing procedure. Specifically, we have eliminated the step of expanding the image edges to 2048×5632 and filling the peripheral regions with 300 K. Instead, we now explicitly incorporate zonal periodicity during the tiling phase. When generating sub-blocks for model input, any tile that crosses the dateline (180°W/180°E) is seamlessly wrapped by incorporating data from the opposite side of the domain. This ensures that the model is never exposed to hard, non-physical boundaries at the dateline during either training or inference.

- L133: Here, it is mentioned that 'spurious MCS' are to be excluded. What are spurious MCS, and if they are still MCS, why should they be excluded?

Response: We thank the reviewer for this comment. The term "spurious MCSs" was indeed potentially misleading, and we agree that it required a more precise explanation. Our intention was to convey that an overly lenient threshold (e.g., 250 K) would identify a large number of cold cloud regions, but not all of these regions represent robust, long-lived MCSs as defined by our study's objectives. These non-robust systems, which we previously referred to as "spurious". To address this comment and improve the clarity of our manuscript, we have revised the relevant sentence in the "Data Processing". The change shifts the focus from excluding "spurious MCSs" to the balanced selection of thresholds to minimize both false negatives and false positives.

- COD vs SOD: I am not completely sure whether I understand the distinction between the two? In Figure 1, it is written that the MCS in tropical and extratropical regions (Region 1 and 2 in the figure) are different in structure? Is it that the MCS differ in their degree of spatial clustering, or do all the individual MCS differ in their structure? For readers less familiar with the distinction between tropical and extratropical MCS some further background information (and references) could be helpful?

Response: We sincerely thank you for raising these critical points. We have thoroughly revised the manuscript to provide a clearer explanation. The difference is indeed structural at the individual MCSs level, stemming from their distinct formative environments. Tropical MCSs, developing in a more uniform environment, tend to have a coherent and compact structure (like a well-defined "blob"). Extratropical MCSs, often embedded in frontal systems, exhibit a more amorphous and diffuse structure, where the intense convective cores are embedded within a larger, less convective cloud shield. Based on the above, we have refined our analogy: Identifying the compact, well-defined tropical MCSs is analogous to Salient Object Detection (SOD), where the target is the most prominent and distinct object in the image. Identifying the embedded convective cores within the diffuse extratropical cloud shield is analogous to Camouflaged Object Detection (COD), where the target is visually similar to its background and lacks clear boundaries. As suggested, we have added a sentence acknowledging the meteorological drivers (e.g., vertical wind shear) behind these structural differences and have included relevant references (e.g., Galarneau Jr et al. (2023);

Muetzelfeldt et al. (2025); Paul et al. (2025)) to provide further background for readers. We thank the reviewer again for helping us strengthen this part of our work.

- Most likely related to the previous point: The title of Section 2.2 is somewhat 'misleading'. I would not have doubted that machine learning can be used to identify MCS. The real point of Section 2.2 is that the authors suggest to use different approaches (SOD vs. COD) for low and mid latitudes. This should be reflected also in the title of the Section 2.2.

Response: We thank the reviewer for this suggestion. Following the reviewer's advice, we have changed the title of Section 2.2 to " The Latitudinal Challenge in Deep Learning-Based MCS Identification" to better reflect our core proposition of employing distinct strategies (SOD for tropics, COD for extratropics) for MCS identification across different latitudinal regions.

- L67-76: This part is less about MCS identification, but more about background information (occurrence frequency, link to precipitation). I think it would better fit into the introduction or, possibly, the discussion.

Response: We thank the reviewer for this suggestion. As the reviewer rightly pointed out, the core theme of this section should remain focused on the identification methodology itself. Following the reviewer's advice, we have evaluated the option of moving this paragraph. However, we found that the connection between MCSs and precipitation is already established in the introduction to provide motivation for the study, and the analysis of this link is further elaborated in the discussion section of our revised manuscript. Therefore, to maintain the conciseness and thematic coherence of the methodology section, we have decided to delete this paragraph (Lines 67-76) entirely. We believe this change significantly improves the focus and clarity of the section. Thank you for the valuable comment.

---

## Author Comment (AC3)

Review of egusphere-2025-3622
Title: MCSeg (v1.0): A Deep Learning Framework for Long-Term Large-Scale Mesoscale Convective
Systems Identification and Precipitation Event Analysis
Aurhors: Peng Li, Zhanao Huang, Yongqiang Yu, Xi Wu, Xiaomeng Huang, and Xiaojie Li

The manuscript describes a deep learning (DL)-based methodology to identify Mesoscale Convective Systems (MCS). The methodology accounts for one single algorithm with different approaches for mid- and low-latitude regions. The performances of the algorithm are compared with other machine learning (ML)-based approaches and with a physical-based approach. The analysis shows comparable results between the new developed algorithm and the physical-based approach, while a general outperformance with respect to the other ML approaches.

The paper is well organized and well written. Although I appreciate the work done by the authors with the development of one single algorithm for both medium and low latitudes, I do not see any particular improvement for the scientific community. As I stated below in the specific comment, an ideal case with perfectly working DL algorithm can exactly reproduce the training and test dataset. In the specific case, the only reason to consider the present work as an added value is the computing time. But, I do not see how a simple threshold-based method can be 200 times slower than a DL based method (hours vs less than two minutes).

**Reply to Referee comment 1:**
- the characteristics of the MCS have to be reported. Not all the readers are familiar with this precipitating structure.

Response: Thank you for your comment. The definition of MCSs, key morphological characteristics, typical life history and its significance in weather and climate have been added at the beginning of the introduction to help non-professional readers understand.

- line 17 and many others: in sentences like this the correct way to report the reference is to put it all in parentheses: (Schumacher and Jonson, 2006).

Response: We have reviewed and corrected the citation format of the references throughout the text to ensure that all citations similar to "(Schumacher and Jonson, 2006)" are placed in parentheses.

- lines 125-126: the use of IMERG final product is only related to the analysis of the precipitation distribution linked to the MCSs carried out in Section 6 and 7? It is not totally clear from the text.

Response: Thank you for your comment. The original text is not clear enough. As we have stated in Section 3.1 "Data information": IMERG precipitation data is only used for the analysis and statistics of precipitation events associated with the identified MCSS.

- lines 137-139: explain better the area coverage threshold. It is not clear whether there must be spatial continuity up to reach 5000 km2 or not. This affect also the interpretation of the results.

When you show, in Figure 6 or 8 for instance, the green and read areas, how these affect the identification of a MCS? I mean, is there a threshold in number of pixels (and consequently in extension area) to say if a given structure is or not a MCS?

Response: Thank you for raising this key question. During the label production stage: The "real" labels we use to train the model are the result of strictly applying physical thresholds. Specifically, a region must meet spatial continuity and its continuous area must exceed 5,000 square kilometers before it can be marked as an MCSs. Therefore, from the very beginning of the model's training, the learning objective has been this "thresholding filtered" MCSs form.

In the model prediction stage: MCSeg is an end-to-end deep learning model. Through learning, the model has internalized the two key physical criteria of "spatial continuity" and "area threshold" into its network parameters. During the training process, the model learned to distinguish between these two types of features by coming into contact with a large number of positive samples (continuous regions with an area greater than 5000 km²) and negative samples (smaller or discontinuous convective regions). Therefore, when the model makes inferences about a new set of data, what it directly outputs are the regions that it "believes" conform to the physical definition of MCSs.

Explanation of the colors in Figure 6/8: The white area represents the MCSs region correctly recognized by the model, that is, the part where the model predicts MCSs and overlaps with the label region defined by the strict physical threshold. The red and green areas indicate the missed recognition and over-recognition of the model. That is, the area predicted by the model as MCSs, but not covered by strict physical threshold labels. This reveals the systematic differences or uncertainties between the "MCSs features" learned by the model and the original physical threshold definitions. For instance, the model might also identify some convective tissues that are close to but slightly below the strict area threshold and have similar physical structures, or the more diffuse parts of the MCSs edge as MCS.

- lines 146-148: what the reason to increase so much the image size? I can understand that at the edges of your image you can suffer of padding, but in Section 4 you mention you use a 3x3 convolutional kernel.

Response: Thank you for this important technical question regarding the input image size. The enlargement was not primarily due to padding from the $3 \times 3$ convolutional kernel, but was a deliberate preprocessing step to address the geographical dimensions of our raw data and to improve the efficiency and coverage of our data sampling during model training.

Our raw satellite brightness temperature data has a global longitudinal coverage of 1715 pixels and a latitudinal coverage of 5143 pixels. To train the model effectively, we divided this global map into smaller, manageable patches. We chose a patch size of $512 \times 512$ pixels as it aligns with common practices in deep learning for segmentation tasks and fits well within GPU memory constraints. However, because 1715 is not an integer multiple of 512, simply tiling the longitudinal dimension would result in underutilization of data near the longitudinal boundaries.

To avoid losing valid data, we applied:

Cyclic padding in longitude: Since the Earth's longitude is periodic, we padded the data along the longitudinal axis to make the total width an integer multiple of 512. This allowed us to create complete $512 \times 512$ patches without discarding any longitudinal data.

Non-cyclic padding in latitude: As latitude is not periodic, we applied padding at the northernmost and southernmost edges to align the latitudinal dimension with the patch size. These padded pixels were filled with a constant background brightness temperature value of 300 K

- Table 1 and 2: I do not understand the choice of multiplying the coefficient by 10. Also because at lines 259-260 you state that all metrics, except M, have 1 as perfect score.

Response: This is a negligence in our expression. To more clearly display the subtle differences of these assessment indicators, we uniformly multiplied them by 10 for display. We add explanations at the corresponding positions in the main text, emphasizing that the perfect score corresponds to 10, not 1, to avoid misunderstandings.

- lines 309-314: really the application of a threshold on 240 images (8 images per day for 30 days in a month) requires almost 3 hours? Honestly, it is hard to trust on this. On the other hand, this does not imply that your approach cannot be faster than the threshold-based one. In addition, Section 5 describes the comparison between MCSeg and other DL-based methods, but your conclusion mainly focus on the shorter computational time with respect to the threshold-based method. Another weak point is the choice of the 10 DL-based methods. At line 261 you state, "We selected 10 comparative models designed for various segmentation task". Basically, you are comparing an MCS oriented method with other "generic" methods. In my opinion, the comparison is not fair.

Response: Thank you for raising this reasonable concern about the reported computation time. We understand that the figure may seem unexpected at first. The stated three-hour processing time is the total time needed for our traditional, CPU-based threshold method to generate MCSS labels for one month of global, high-resolution satellite data. This process is not only a simple pixel-level thresholding. It also includes: identifying all connected cold-cloud regions in each image, accurately calculating the real area of each region, and finally selecting only those regions larger than 5,000 km² as valid MCSs. All these steps are done in sequence without special optimization, which makes it slow for processing such large and detailed global data. In comparison, after about ten hours of one-time offline training, our proposed MCSeg deep learning model uses GPU acceleration to process approximately 130 global images per minute. This represents a major improvement in speed.

Regarding the choice of models for comparison, we agree that there are currently no publicly available deep learning models made specifically for MCSS recognition to compare with directly. Therefore, we trained several general-purpose image segmentation models on the same dataset as a baseline. Our goal is not to make a perfectly equal comparison, but to show that a model like MCSeg, which is designed with knowledge of MCSs characteristics, can perform better on this specific scientific task than general models.

We are committed to reproducibility. The code for the threshold method is adapted from Huang et al. (their code and data are public). Our MCSeg model code, trained weights, and the dataset we used are also publicly available. We encourage others to test and verify both the processing speeds and model performance independently, as openness is essential to our research.

- lines 323-324: I disagree with this sentence. If your model was perfect, it would be able to exactly reproduce the training (and the test, supposing perfect generalization capabilities) data. This means that you are not able to enhance the threshold-based method but, at most, to equal it (since your dataset is built exploiting the threshold-based method).

Response: Your theoretical understanding is completely correct. The upper limit of a "perfect" DL model in an ideal situation is to reproduce the label (i.e., the result of the threshold method). However, the actual goal of MCSeg is not to reach this theoretical upper limit, but to achieve computational acceleration while maintaining a high precision comparable to that of the threshold method, thereby making large-scale MCSs feature analysis possible.

- the last part of the paper (Figure 9-12 and Section 7) could be considered a bit out of topic with respect to the rest of the paper. The analysis is useful (no doubt), but cannot be considered a consequence of the development of the MCSeg algorithm. In particular, the results shown in Figure 9 highlight negligible differences between MCSeg and threshold-based algorithms. In addition, vertical and horizontal bands (lines 338-339 – I would add curving bands as well) present in Figure 10 have to be fixed. A data post-processing can be applied in order to remove this issue.

Response: Thank you for your insightful feedback. We agree that it is necessary to clarify and strengthen the connection between algorithm development and subsequent scientific analysis. In response, we made extensive revisions to better integrate these sections into the core narrative of the paper.

To make the connection between the chapters of the paper tighter, we merged the last two chapters of the paper, and now the title is " Comprehensive Validation of the MCSeg Algorithm ". We have redefined comparative analysis as a crucial, climate-focused verification step. The main purpose is to demonstrate that our deep learning model MCSeg can generate statistically and climate-consistent MCSS recognition compared with the physical threshold methods established globally. For the vertical/horizontal strip problem. After the MCSS recognition, we processed it with a strip-removing filter. The revised figure in the manuscript now shows a clean one without any stripes.